# OH and HO$_2$ radical chemistry at a suburban site during the EXPLORE-YRD campaign in 2018

Xuefei Ma[1], Zhaofeng Tan[2], Keding Lu[1,*], Xinping Yang[1], Xiaorui Chen[1], Haichao Wang[1,3],
Shiyi Chen[1], Xin Fang[1], Shule Li[1], Xin Li[1], Jingwei Liu[1], Ying Liu[1], Shengrong Lou[4], Wanyi
Qiu[1], Hongli Wang[4], Limin Zeng[1], Yuanhang Zhang[1,5,6,*]

[1]State Key Joint Laboratory of Environmental Simulation and Pollution Control, College of Environmental
Sciences and Engineering, Peking University, Beijing, China

[2]Institute of Energy and Climate Research, IEK-8: Troposphere, Forschungszentrum Juelich GmbH, Juelich,
Germany

[3]School of Atmospheric Sciences, Sun Yat-sen University, Guangzhou, China

[4]State Environmental Protection Key Laboratory of Formation and Prevention of the Urban Air Complex,
Shanghai Academy of Environmental Sciences, Shanghai, China

[5]Beijing Innovation Center for Engineering Sciences and Advanced Technology, Peking University, Beijing,
China

[6]CAS Center for Excellence in Regional Atmospheric Environment, Chinese Academy of Science, Xiamen, China

*Correspondence to*: K. Lu (k.lu@pku.edu.cn), Y. Zhang (yhzhang@pku.edu.cn)

## Abstract

The first OH and HO$_2$ radical observation in Yangtze River Delta, one of the four major urban agglomerations
in China, was carried out at a suburban site Taizhou in summer 2018 from May to June, aiming to elucidate
the atmospheric oxidation capacity in this region. The maximum diurnal averaged OH and HO$_2$
concentrations were $1.0\times10^7$ cm$^{-3}$ and $1.1\times10^9$ cm$^{-3}$, respectively, which were the second highest HO$_x$ (sum
of OH and HO$_2$) radical concentrations observed in China. HONO photolysis was the dominant radical
primary source, accounting for 42% of the total radical initiation rate. Other contributions were from

carbonyl photolysis (including HCHO, 24%), $O_3$ photolysis (17%), alkenes ozonolysis (14%), and $NO_3$
oxidation (3%). A chemical box model based on RACM2-LIM1 mechanism could generally reproduce the
observed $HO_x$ radicals, but systematic discrepancy remained in the afternoon for OH radical, when NO
mixing ratio was less than 0.3 ppb. Additional recycling mechanism equivalent to 100 ppt NO was capable
to fill the gap. The sum of monoterpenes was on average up to 0.4 ppb during daytime, which was allocated
all to $\alpha$-pinene in the base model. Sensitivity test without monoterpene input showed the modelled OH and
$HO_2$ concentrations would increase by 7% and 4%, respectively, but modelled $RO_2$ concentration would
significantly decrease by 23%, indicating that monoterpene was an important precursor of $RO_2$ radicals in
this study. Consequently, the daily integrated net ozone production would reduce by 6.3 ppb if without
monoterpene input, proving the significant role of monoterpene on the photochemical $O_3$ production in this
study. Besides, the generally good agreement between observed and modelled $HO_x$ concentrations suggested
no significant $HO_2$ heterogeneous uptake process during this campaign. Incorporation of $HO_2$ heterogeneous
uptake process would worsen the agreement between $HO_x$ radical observation and simulation, and the
discrepancy would be beyond the measurement-model combined uncertainties using an effective uptake
coefficient of 0.2. Finally, the ozone production efficiency (OPE) was only 1.7 in this study, a few folds
lower than other studies in (sub)urban environments. The low OPE indicated a slow radical propagation rate
and short chain length. As a consequence, ozone formation was suppressed by the low NO concentration in
this study.

## 1. Introduction

Stringent air quality regulations have been implemented in China for more than a decade to combat the
severe air pollution problems, and dramatic reduction of primary air pollutants such as sulfur dioxide ($SO_2$),
nitrogen oxides ($NO_x$), and coarse particulate matters ($PM_{10}$) has achieved. Besides, a significant decrease
in fine particulate matters ($PM_{2.5}$) is found since 2013, when the Chinese government took the strictest
measures to reduce the anthropogenic emission in the polluted regions (Wang et al., 2020b;Wang et al.,
2019b). However, the surface ozone ($O_3$) showed a contrasting trend with an increasing rate of 1-3 ppb $a^{-1}$
over the Chinese eastern megacity clusters, among which North China Plain and Yangtze River Delta regions
are of the most significant increase of 3-12 ppb $a^{-1}$ (Wang et al., 2020b). The only known formation pathway
to $O_3$ in the troposphere is the photolysis of $NO_2$ (R1 and R2). The increasing $O_3$ despite the successful
reduction in $NO_2$ demonstrates the nonlinearity of the photochemistry caused by the dual role of $NO_x$.
$NO_2 + h\nu \rightarrow NO + O(^3P)$ ($\lambda < 398$ nm)          (R1)
$O(^3P) + O_2 + M \rightarrow O_3 + M$          (R2)
The ozone formation nonlinearity can be described by investigating $HO_x$ radical chemistry (Tan et al.,
2018a;Tan et al., 2018b). In low $NO_x$ conditions, the local ozone production rate $P(O_3)$ increases with $NO_x$
due to the efficient NO to $NO_2$ conversion by peroxy radicals (R3-R4). In high $NO_x$ conditions, $P(O_3)$
decreases with $NO_x$ because the radical termination (R5) overwhelms the radical propagation processes. The
key is to find the optimized reduction strategy for both $NO_x$ and VOCs to efficiently control the $O_3$ production,
which the radical measurement could give insight to.
$HO_2 + NO \rightarrow OH + NO_2$          (R3)
$RO_2 + NO \rightarrow RO + NO_2$          (R4)
$OH + NO_2 \rightarrow HNO_3$          (R5)
Numerous field campaigns focusing on the hydroxyl (OH) and hydroperoxy radical ($HO_2$) measurements
have been performed worldwide for the past decades, covering various environments including forest,
marine, remote, polar, rural, suburban, and urban (Stone et al., 2012). The measured OH concentrations
varied in an order of magnitude (in the range of $10^6$-$10^7$ $cm^{-3}$) among different types of environments, and
the OH daily maximum concentrations showed a tendency of higher values in urban areas. Six field
campaigns have been implemented in China during summer periods, namely the Backgarden (2006), Heshan
(2014), Shenzhen (2018) campaigns in Pearl River Delta (PRD) (Lu et al., 2012;Tan et al., 2019a;Wang et
al., 2019a), and Yufa (2006), Wangdu (2014), and Beijing (2017) campaigns in North China Plain (NCP)
(Lu et al., 2013;Tan et al., 2017;Whalley et al., 2021) to investigate the atmospheric oxidation capacities and
photochemistry characteristics of two of the most polluted regions in China, in which Backgarden campaign
reported the highest OH concentration ($15\times10^6$ $cm^{-3}$) ever observed (Lu et al., 2019). Chemical box model
simulation based on conventional mechanisms could generally reproduce the OH radical concentrations in
these Chinese campaigns at NO concentration above 1 ppb, but a tendency to underestimate OH radical are
continuously observed at NO concentration less than 1 ppb, which is a common feature in isoprene-rich
forest environments and OH concentration could be underestimated by a factor of up to 10 (Rohrer et al.,
2014;Tan et al., 2001;Lelieveld et al., 2008). Novel recycling mechanism related to isoprene and its
degradation products without the involvement of NO has been considered as a possible reason for the OH
measurement-model discrepancy in isoprene-rich environments (Peeters et al., 2009;Peeters et al.,
2014;Lelieveld et al., 2008), but it is not sufficient to explain the large discrepancy for campaigns in urban
and suburban environments. Moreover, even in isoprene-rich environments, the inclusion of the novel
recycling mechanism of isoprene is still not sufficient to reproduce the observed OH concentrations (Stone
et al., 2011b). It is worth noting that the high OH concentration might be caused by an unknown interference
in OH measurements by laser induced fluorescence (LIF) (Mao et al., 2012;Novelli et al., 2014;Hens et al.,
2014;Feiner et al., 2016). Mao et al. (2012) reported that up to 80% of OH measurement is interference in a
pine forest. However, the interference was minimal and within the instrumental detection limit in other
campaigns under urban and suburban environments by different LIF instruments (Griffith et al., 2016;Tan et
al., 2017;Woodward-Massey et al., 2020). Therefore, the OH measurement accuracy needs to be addressed
prior to critical discussion about defects in our knowledge of the radical chemistry.
Yangtze River Delta (YRD) region is one of the four major polluted regions in China and $O_3$ has become the
most critical pollutant in this region (Li et al., 2019). A four-year continuous observation showed the ozone
pollution days have more than doubled from 2014 to 2017 (28 to 76 days) in YRD region (Liu et al., 2020b).
Lu et al. (2018) reported that the monthly averaged daily maximum 8-h concentrations of $O_3$ were even
higher in YRD than in the NCP. Plenty of studies have been performed to investigate the ozone pollution
characteristics and diagnose the sensitivity of ozone formation to its precursors over this region (Zhang et
al., 2020;Ding et al., 2013;Tie et al., 2013;Geng et al., 2015;Xing et al., 2017), but none of the studies were
deployed with $HO_x$ radical observations. In the present study, we report a new radical observation in YRD
region during the campaign EXPLORE-YRD (EXPeriment on the eLucidation of the atmospheric Oxidation
capacity and aerosol foRmation, and their Effects in Yangtze River Delta) together with a comprehensive set
of trace gases measurements. It provides a unique chance to investigate the photochemistry with the support
of $HO_x$ radical observation in this region. Besides, the in-situ $HO_x$ radical observation also allows to
investigate the impact of potential mechanisms such as $HO_2$ heterogeneous uptake on the photochemistry.

## 2. Methodology

## 2.1 Measurement site

The EXPLORE-YRD campaign was conducted in the summer of 2018 (14 May to 20 June) in the park of meteorological radar station in suburban Taizhou (32.56˚N, 119.99˚E), Jiangsu Province, which is approximately 200 km north-west and 100 km north-east of the two major megacities, Shanghai and Nanjing, in Yangtze River Delta region (Fig. S1). The site was surrounded by fishponds and grass lands, featured with strong biogenic emission and occasionally biomass burning. No major industrial emissions were found within 500 meters. The closest road with slight traffic was about 100 meters to the South, and to the North and East of the measurement site were the highways S28 and S35 with moderate traffic. For most of the campaign, southerly and easterly winds prevailed, and brought air from the megacities and sea in upwind to this site during the daytime. Thus, the sampled air mass during this campaign could generally embody the atmospheric chemical characteristics in this region.

## 2.2 OH and HO$_2$ radical measurements

OH and HO$_2$ radicals were measured by the Peking University Laser Induced Fluorescence system (called PKU-LIF), which was successfully deployed several times in previous campaigns in Pearl River Delta and North China Plain regions in China (Tan et al., 2017;Tan et al., 2018c;Tan et al., 2019a;Ma et al., 2019). OH radical is detected by laser-induced fluorescence at a low pressure cell (4 $h$Pa) after a sampling nozzle (Hofzumahaus et al., 1998;Holland et al., 2003). The OH signal is determined by tuning the laser wavelength (308 nm) on- and off-line, so-called wavelength modulation. Specific description of the instrument configuration could be found in (Tan et al., 2017) and references therein.

HO$_2$ radical is chemically converted to OH by reaction with NO that is injected into the flow through a ring-shaped injector installed below the sampling nozzle and then is detected in the form of OH in the second detection cell. Previous studies indicated that part of the RO$_2$ species derived from longer chain alkanes (> C3), alkenes, and aromatic compounds (namely complex-RO$_2$) have the potential to rapidly convert to OH on the same time scale as HO$_2$ inside the fluorescence cell, and thus, might cause interference for HO$_2$ measurement (Fuchs et al., 2011;Whalley et al., 2013). To minimize the potential interference from RO$_2$, the

added NO mixing ratio was switched between 2.5 ppm and 5 ppm every 2 minutes, corresponding to the
$HO_2$ conversion efficiencies of 10% and 20%, respectively. The expected $RO_2$ conversion efficiency for both
modes was below 10% for this experimental setup for isoprene derived $RO_2$ from laboratory tests (Fuchs et
al. 2011). The extent of the $RO_2$-interference was also proportional to the complex-$RO_2$-to-$HO_2$ ratio.
Unfortunately, $RO_2$ was not measured during this campaign but one would expect a strong correlation
between $RO_2$ (or complex-$RO_2$) and $HO_2$ (Tan et al., 2017; Whalley et al., 2021). Previous field summer
campaigns in China showed that, the ratio of complex-$RO_2$ to $HO_2$ varies from 0.6 at a rural site in Wangdu
(Tan et al., 2017) to 2 at an urban site in Beijing (Whalley et al., 2021). As the chemical condition
encountered in YRD was more similar to that of Wangdu (the Beijing campaign was conducted at an urban
site), it was reasonable to assume the complex-$RO_2$ to $HO_2$ ratio in this study was closer to 0.6. Therefore,
by applying the $RO_2$ conversion efficiency of 0.1 as an upper limit, the maximum $HO_2$ interference from
$RO_2$ radicals should be closer to 6% of the $HO_2$ measurement in this study assuming complex-$RO_2$ to $HO_2$
ratio to be 0.6.
The PKU-LIF instrument was calibrated every 2 days during the campaign using a radical calibration source
(Hofzumahaus et al., 1996;Holland et al., 1998). Stable sensitivities were found over the whole campaign
with reproducibility of 1.2% and 8.0% for OH and $HO_2$, respectively ($1\sigma$ standard deviation). Thus, averaged
sensitivity was applied for the radical concentration determination. Considering the combined uncertainty of
calibration source (10%, $1\sigma$) with reproducibility of calibrated sensitivities, the accuracies of OH and $HO_2$
measurement were 10% and 13%, respectively. The detection limits of OH and $HO_2$ measurements using
LIF technique depend on the sensitivity, the laser power, the background signal, and the integration time
(Holland et al., 1995), and were $6.0\times10^5$ cm$^{-3}$ for OH and $1.0\times10^7$ cm$^{-3}$ for $HO_2$ at a typical laser power of
12 mW for a data acquisition time of 30 s (for signal-to-noise ratio of 2).
Several studies conducted in forested environments indicated that OH measurements by Laser-Induced
Fluorescence technique using wavelength modulation method might suffer from unknown internal-produced
interference (Mao et al., 2012;Novelli et al., 2017), and the magnitude of interference is highly dependent
on the specific design of the instrument, the operating parameters, and the type of environment in which the
instrument is deployed (Fuchs et al., 2016;Novelli et al., 2014;Woodward-Massey et al., 2020;Cho et al.,
2021). To investigate the possible OH interference in this campaign, we performed an extended chemical
modulation experiment on 7 June. During the experiment, a chemical modulation device consisting of a
Teflon tube with an inner diameter of 1.0 cm and a length of 10 cm was placed on the top of the OH sampling
nozzle. About 17 slpm (standard liter per minute) of ambient air was drawn through the tube by a blower, 1
slpm of which entered the fluorescence cell. Tests on the transmission efficiency of OH through the chemical
modulation device showed that the signals differed by less than 7% with or without chemical modulation
device, indicating the losses of ambient OH to the chemical modulation device were insignificant. For
ambient measurement application, either propane (a 12% mixture in nitrogen, 6 sccm) diluted in a carrier
flow of pure nitrogen (200 sccm) or pure nitrogen (200 sccm) was injected into the center of the tube
alternatively every 5 minutes via two oppositely posited needles at the entrance of Teflon tube. The ambient
OH signal can be then deduced by differentiating the signals from adjacent measurement modes with and
without propane injection. The amount of the scavenger added is typically selected to be sufficiently high
for reacting with ambient OH but not in excess in case reacting with internal-produced OH, and thus, the
scavenging efficiency is usually kept around 90%. Calibrations of OH sensitivity with and without propane
injection showed the scavenging efficiency of OH was around 93% in this experiment, and the kinetic
calculation indicated the added propane removed less than 0.7% of the internal-produced OH. Therefore, the
real ambient OH concentration can be obtained by multiplying the differential OH signal by the scavenging
efficiency and by the instrument sensitivity. More details about the prototype chemical-modulation reactor
used with PKU-LIF and the calculation method can be seen in Tan et al. (2017).
**2.3 Trace gases measurements**
A large number of trace gases and aerosol properties related to the atmospheric oxidation chemistry
investigation were measured simultaneously. Instruments were placed in sea-containers with their sampling
inlets mounted 5 meters above ground. The detail of instrumentation is described by (Wang et al., 2020a).
In Table 1, the measured species related to photochemistry study are listed together with the performance of
instruments.
$O_3$, NO, $NO_2$, $SO_2$ and CO were detected by a series of commercial analyzers from Thermo Inc. $O_3$ was
measured by a UV Photometric analyzer (Model 49i). Both NO and $NO_2$ were measured by a trace-level
analyzer (Model 42i) using chemiluminescence method. Therein, $NO_2$ measurement was accomplished by a
home-built photolytic converter to avoid interference from other $NO_y$ species. HONO measurement was
deployed by a Long-path Absorption Photometry with a time resolution of 1 min. A gas chromatograph
coupled with a flame ionization detector and mass spectrometer (GC-FID-MS) was deployed to measure
volatile organic compounds (VOC) including non-methane hydrocarbons (C2-C11 alkanes, C2-C6 alkenes,
C6-C10 aromatics, isoprene, sum of monoterpenes), and oxygenated VOCs including methyl vinyl ketone
(MVK)/Methacrolein (MACR), methyl-ethyl-ketone (MEK), acetaldehyde (ACD), acetone (ACT) in a time
resolution of 1 hour. The sum of monoterpenes was also detected by proton transfer reaction mass
spectrometry (PTR-MS). Formaldehyde and glyoxal were measured by a commercial and a home-built
instruments, namely Hantzsch and CEAS, respectively. Additionally, meteorological parameters including
temperature, relative humidity, pressure, wind speed, and wind direction were all measured simultaneously.
Photolysis Frequencies was calculated by integrated actinic flux measured by a spectroradiometer.

## 200     2.4 Model description

An observation-constrained box model based on RACM2-LIM1 mechanism (Goliff et al., 2013;Peeters et
al., 2014) was used to simulate the OH and $HO_2$ radical concentrations. Briefly, observations of the
photolysis frequencies $j(O^1D)$, $j(NO_2)$, $j(HONO)$, $j(H_2O_2)$, $j(HCHO)$, and $j(NO_3)$, $O_3$, NO, $NO_2$, CO, $CH_4$,
$SO_2$, HONO, C2-C12 VOCs, and certain oxygenated VOCs such as HCHO, acetaldehyde, glyoxal and
acetone as well as the meteorological parameters were used to constrain the model with a time resolution of
5 min. Photolysis frequencies of other species were calculated in the model using the following function of
solar zenith angle ($\chi$) and scaled to the ratio of measured to calculated $j(NO_2)$ to represent the effect from
clouds. :
$J = l \times (\cos\chi)^m \times e^{-n \times \sec\chi}$        (Eq. 1)
where the optimal values of parameters $l$, $m$, and $n$ for each photolysis frequency were adopted (Saunders et
al., 2003). The organic compounds were not treated individually but assigned to different lumped species
according to the reactivities with OH. The classification of the constrained organic compounds in RACM2
were listed in Table 2 in detail. The sum of monoterpene was allocated to $\alpha$-pinene in the model and the
uncertainty due to such simplification was discussed in Sect. 4.2.2. Isomerization of isoprene-derived peroxy
radicals was also considered. Other lumped secondary species were unconstrained due to the technical limits
but generated numerically by the model calculation.
Additional first-order loss term equivalent to a lifetime of 8 hours was given to all species to represent
physical losses by means of deposition, convection, and advection. The observed-to-model ratio of PAN
concentration was 1.09 using this physical loss rate, while the modelled PAN concentration agreed to
measurements from late morning to the midnight but slightly lower than measurements in the early morning
(Fig. S2), which might be related to the effect of boundary layer height variation. To test the influence of
boundary layer height diurnal variation, we performed a sensitivity test by imposing a boundary layer height
(BLH, reanalysis data from European Centre for Medium-Range Weather Forecasts) dependent loss rate to
all species. In this scenario, the model continuously underpredicted the concentration in the early morning,
and additionally, the model overestimated the observed PAN in the midday and afternoon (Fig. S2). This
was because the boundary layer height dependent loss rate was largest at night, which made the loss of PAN
greater and further worsened the measurement-model comparison. Therefore, the treatment of a first-order
loss term equal to 8 hours to all species in the model might not reflect the loss due to deposition but gave a
reasonable approximation on the overall physical loss of the model-generated intermediates. Nevertheless,
the modelled OH and $HO_2$ concentrations were insensitive to the imposed loss rate (Fig. S3). The
concentrations differed less than 0.5% between two cases for both OH and $HO_2$. In addition, sensitivity test
without HCHO and glyoxal constrained indicated that model would under-predicted the HCHO and over-
predicted the glyoxal concentrations (Fig. S2), which might be related to the significant primary emission of
HCHO and missing sinks of glyoxal in the current mechanisms. However, the missing sources and sinks of
HCHO and glyoxal are not the scope of this study. To avoid interruption from incapability of model
performance, both HCHO and glyoxal were constrained to observations in this study.
According to the Monte-Carlo simulation tests, the estimated $1\sigma$ uncertainty of the model calculation was
32% and 40% for OH and $HO_2$, respectively, arising mainly from the uncertainties of both observational
constraints and kinetic rate constants, among which the rate constant between $HO_2$ and NO, dilution time
and NO concentration were of most significant importance in this study.

## 3. Results

### 3.1 Meteorological and chemical conditions

The meteorological condition encountered during the campaign was characterized by high temperature (up to 35 ˚C), high relative humidity (54% on average) and strong solar radiation. The wind speed was usually below 2 m s$^{-1}$ during the daytime. Back trajectory analysis demonstrated that the air masses were predominately transported from the South and East during the campaign (Fig. S4). High $O_3$ concentrations were frequently observed on days when the air masses transported to the measurement site had passed through the South especially the Southwest large city clusters. As shown in Fig. 1, the daytime $O_3$ concentrations exceeded the Chinese national air quality standard level II (hourly averaged limit 93 ppb) on several days and reached as high as 150 ppb on 5 and 6 June.

Figure 2 shows mean diurnal profiles of the key parameter observations. The averaged period is selected when $HO_x$ measurements were available (23 May-17 June excluding the break). Solar radiation was intense during the whole campaign indicated by photolysis frequencies $j(O^1D)$ and $j(NO_2)$. NO concentration peaked at 4 ppb during morning rush hour and then dropped to 0.2 ppb at noon. $O_3$ concentration started to increase after sunrise and reached the peak of 86 ppb around noon and lasted until sunset. Subsequently, $O_3$ concentration decreased and partially converted to $NO_2$ due to the absence of sunlight. The total oxidant ($O_x$), the sum of $O_3$ and $NO_2$ also decreased after sunset. Along with the increased $NO_2$ at night, HONO concentration increased and reached the maximum of up to 1.3 ppb at sunrise and then declined rapidly due to the fast photolysis. The averaged HONO concentration was 0.6 ppb on the daytime basis. Peroxyacyl nitrates (PAN) is an indicator for active photochemistry which increased since sunrise reaching maximum of 1.6 ppb at 12:00 and then decreased in late afternoon during this campaign. However, other oxidation products, including HCHO and glyoxal, similar to CO and $SO_2$, peaked at 8:00 CNST rather than in the noon and late afternoon and decreased afterwards, indicating an anthropogenic emission-related origin of these species. Since this campaign was conducted during a harvest season, agriculture biomass burning might be responsible for the elevated HCHO and glyoxal in the early morning (Guo et al., 2021;Liu et al., 2020a;Wang et al., 2017;Silva et al., 2018).

Isoprene showed a broad peak of 0.2 ppb from 09:00 to 15:00, which was several times lower than during

the previous summer campaigns (Lu et al., 2012;Lu et al., 2013;Tan et al., 2017). The sum of monoterpene
concentrations varied from 0.2 ppb to 0.4 ppb showing a diurnal peak around noon. Though the speciation
is not known, the daytime monoterpene concentration was comparable to monoterpene dominated pine forest
(Kim et al., 2013;Hens et al., 2014). The role of monoterpene to $HO_x$ chemistry is discussed in section 4.2.2.

## 3.2 OH and $HO_2$ radical observation

Figure 3 shows the time series of the observed and calculated OH and $HO_2$ radical concentrations.
Continuous measurement of $HO_x$ radicals was interrupted by the rainfalls and calibration or instrument
maintenance. Distinct diurnal variation was observed for both OH and $HO_2$ radical. The daily maxima of OH
and $HO_2$ concentration were in the range of $(8-24)\times10^6$ $cm^{-3}$ and $(4-28)\times10^8$ $cm^{-3}$, respectively. The mean
diurnal profiles showed that averaged OH and $HO_2$ peak concentrations (1-h averaged) were $1.0\times10^7$ $cm^{-3}$
and $1.1\times10^9$ $cm^{-3}$, respectively (Fig. 4). Additionally, the chemical modulation tests performed on 7 June, an
$O_3$ polluted day, indicated the unknown OH interference, if existed, was insignificant and below the detection
limits during this campaign (Fig. S5).
For comparison, the daytime measured OH concentration in this campaign together with the OH
concentrations in Yufa and Wangdu campaigns in NCP region and in Backgarden, Heshan and Shenzhen
campaigns in PRD region, where OH radical observations were available in China were summarized in Table
3 and Figure 5. Overall, the OH radical concentration at present study was relatively higher than during other
campaigns except for the Backgarden campaign in 2006 (Hofzumahaus et al., 2009). A recent winter
observation in Shanghai in YRD region reported an averaged noontime OH concentration of $2.7\times10^6$ $cm^{-3}$
(Zhang et al., 2022), which was comparable to or even higher than that was observed in winter Beijing
$(1.7\sim3.1\times10^6$ $cm^{-3})$ (Tan et al., 2018c;Ma et al., 2019;Slater et al., 2020). It demonstrated the strong
atmospheric oxidation capacity in this region among the three megapolitan areas (NCP, PRD, and YRD) in
China from the perspective of OH concentration.
We also found strong correlation between observed OH radical concentration and photolysis frequency
$(j(O^1D))$ during the EXPLORE-YRD campaign, with the correlation coefficient $R^2$ and the correlation slope
being 0.85 and $4.8\times10^{11}$ s $cm^{-3}$, respectively (Fig. 6). Notably, the slopes were in the range of $(4.0-4.8)\times10^{11}$
s $cm^{-3}$ for all the previous filed campaigns in NCP and PRD regions, for both summer and winter (Tan et al.,
2017;Tan et al., 2018c;Lu et al., 2012;Ma et al., 2019). It suggested that the atmospheric oxidation capacity
to sustain the radical concentrations was comparable under various chemical conditions in the three major
urban agglomerations. Besides, the intercept of the linear fit for this campaign was about $7.6\times10^5$ cm$^{-3}$, which
was comparable to the Wangdu campaign in 2014 ($7.7\times10^5$ cm$^{-3}$) and lower than the Yufa and Backgarden
campaigns in 2006 ($1.6\times10^6$ cm$^{-3}$ and $2.4\times10^6$ cm$^{-3}$, respectively). It represented the non-photolytically
produced OH concentration.

## 3.3 Modelled OH reactivity

OH reactivity ($k_{OH}$) is the pseudo first-order loss rate coefficient of OH radical, and indicates the inverse of
the chemical lifetime of OH radical. It can be defined by the sum of the OH reactants concentrations
multiplied by their reaction rate constants versus OH radical (Fuchs et al., 2017;Yang et al., 2016;Yang et
al., 2019;Lou et al., 2010):
$k_{OH} = \sum_i k_{OH+X_i} [X_i]$                (Eq. 2)
In this study, the $k_{OH}$ was calculated from measured NO, NO$_2$, CO, CH$_4$, SO$_2$, C2-C12 VOCs (including
isoprene and monoterpene), HCHO, acetaldehyde, glyoxal, and acetone and model-generated intermediate
species (mainly referred to the unconstrained oxygenated VOCs). The calculated $k_{OH}$ ranged between 5 s$^{-1}$
and 40 s$^{-1}$ (Fig. 3).
The typical mean diurnal variation of $k_{OH}$ showed a peak in the early morning and then dropped by nearly
50% to a minimum in the afternoon (Fig. 7a). The averaged $k_{OH}$ for periods with OH radical measurement
was 10.8 s$^{-1}$ on daytime basis (08:00-16:00), and a total of 36% of the modelled $k_{OH}$ could be attributed to
the inorganic compounds (Fig. 7b). CO was the single largest contributor to $k_{OH}$, with a campaign average
contribution of 19%. NO and NO$_2$ together contributed 15% of the modelled $k_{OH}$. Alkanes, alkenes, and
aromatics contributed additional 15% of the modelled $k_{OH}$. The reactivity from isoprene made a small
contribution (5%) to the modelled $k_{OH}$ compared to other campaigns conducted in suburban China, where
isoprene typically contributed about 20% of the total $k_{OH}$ (Lou et al., 2010;Fuchs et al., 2017). The
contributions that monoterpene made was 4%, which was a substantial faction considering that the daytime
monoterpene level was usually low in suburban and urban area.
The OVOCs made up a large portion, accounting for approximately 40% of the modelled $k_{OH}$. The model-
generated OVOCs made comparable contribution to the measured ones (22% vs. 18%), and the model-
generated contribution to OH reactivity was insensitive to the imposed physical loss rate (Fig. S3). This
characteristic was similar to what was observed in London and Wangdu (Whalley et al., 2016;Fuchs et al.,
2017), where major OVOCs including HCHO, acetaldehyde, and acetone were directly measured and the
measured OVOCs together with the modeled-generated OVOCs accounted for a large portion of the total
reactivity (44% and 25%, respectively). It was noteworthy that, in both campaigns, $k_{OH}$ was directly
measured and the $k_{OH}$ budget was largely closed. In some previous studies in urban and suburban areas,
however, missing $k_{OH}$ ranging from less than 30% to over 50% of the total reactivity was often observed
(Kovacs et al., 2003;Lou et al., 2010;Shirley et al., 2006;Yang et al., 2016). The common feature of these
observations was that the measurement of OVOCs was completely missing. In fact, model simulations had
proved that the model-generated OVOCs from the photooxidation of measured VOCs could quantitatively
explain the missing $k_{OH}$ in most of these campaigns during daytime, and the majority of the model-generated
OVOCs were HCHO, acetaldehyde, glyoxal, and the isoprene oxidation products. Therefore, in recent
studies, with the improved coverage of the measurement of major OVOCs species, together with the model-
generated secondary species, the calculated $k_{OH}$ was largely in agreement with the measured $k_{OH}$ in urban
and suburban areas during the daytime. However, significant difference could still be observed in areas
affected by dramatic anthropogenic influences, for instance in central Beijing (Whalley et al., 2021), 30% of
the measured $k_{OH}$ remained unaccounted for, even if the measured and model-generated OVOCs were taken
into account, which only contributed 6.5% of the total reactivity, implying that the missing reactivity could
be attributed to the undetected or unrecognized species under complex environments.
**4. Discussion**
**4.1 Sources and sinks of RO$_x$ radicals**
The sum of OH, HO$_2$, and RO$_2$ radicals are known as RO$_x$ radical. The interconversion within the RO$_x$ radical
family is relatively efficient via radical propagation reactions, in which the number of consumed and
produced radicals are equal and do not change the total RO$_x$ concentrations. In this section, we concentrate
on the radical initiation processes that produce radicals from non-radical molecules, and chain termination
processes that destroy radicals. The radical primary production consists of photolysis reactions and alkene
ozonolysis. Radical termination processes include reactions with nitrogen oxides and recombination of
peroxy radicals.
Figure 8 presents the mean diurnal profiles of $RO_x$ radical production and destruction rates based on the
model calculation. The $P(RO_x)$ and $L(RO_x)$ show distinct diurnal variation with maximum of 6.8 ppb $h^{-1}$ at
noontime. In other campaigns (Table 3), diurnal maximum $P(RO_x)$ varies from 1.1 ppb $h^{-1}$ at a suburban site
in Nashville to about 11.6 ppb $h^{-1}$ at a rural site near London during a heatwave (Martinez, 2003;Emmerson
et al., 2007). The $P(RO_x)$ in EXPLORE-YRD campaign is comparable to those found in Mexico 2003,
Mexico 2006 and Yufa 2006 (Mao et al., 2010;Dusanter et al., 2009b;Lu et al., 2013) .
The daytime averaged radical chemistry production rate was 5.7 ppb $h^{-1}$, of which 83% was attributed to
photolytic process. HONO photolysis was the dominant primary source for the entire day and contributed
up to 42% of $P(RO_x)$ on daytime basis. Two recent winter campaigns in the same region also found HONO
photolysis dominated radical primary source, contributing 38% to 53% of the total radical sources, despite
the overall radical production rates were several times lower than that in summertime (Lou et al., 2022;Zhang
et al., 2022). In fact, the photolysis of HONO is one of the most important radical primary sources in
worldwide urban and suburban areas for both summer (Ren et al., 2003b;Dusanter et al., 2009b;Michoud et
al., 2012;Whalley et al., 2018;Tan et al., 2017) and winter time (Ren et al., 2006;Kanaya et al., 2007;Kim et
al., 2014;Tan et al., 2018c;Ma et al., 2019). Besides, carbonyl compounds (including HCHO) photolysis was
also an important contributor to radical primary sources under urban and suburban conditions (Kanaya et al.,
2007;Griffith et al., 2016;Emmerson et al., 2007). In this study, carbonyl compounds photolysis accounted
for on average 24% of $P(RO_x)$, in which 14% was from HCHO solely. The dominant primary radical source
in remote regions, ozone photolysis (generating $O^1D$ and subsequently reacts with $H_2O$ to produce OH), also
played a significant role in this study, contributing 17% to $P(RO_x)$. Besides, the non-photolytic radical source
alkene ozonolysis peaked at around 10:00 in the morning, and the most important $O_3$ reactant was
monoterpene (35% on daytime basis). It was worth noting that $P(RO_x)$ reduced significantly after sunset
while there was a small peak of 1.5 ppb $h^{-1}$ appeared at dusk. The nighttime radical chemistry was mainly
initiated by $NO_3$ oxidation (82%) with monoterpene in the first half of the night, but the $NO_3$ chemistry was
suppressed from midnight to sunrise by the increasing NO concentration because of the efficient titration
effect (Wang et al., 2020a).
During the EXPLORE-YRD campaign, the $RO_x$ termination processes were mainly dominated by the
$OH+NO_2$ reaction before 08:00 and by peroxy radical self-reaction in the afternoon (Fig. 8). On daytime
basis, nitrate formation and peroxy radical recombination both accounted for half of $L(RO_x)$. The peroxy
radical recombination including $HO_2+RO_2$, $HO_2+HO_2$, and $RO_2+RO_2$ reactions contributed 33%, 15%, and
1% to $L(RO_x)$, respectively. Because the $HO_2$ and $RO_2$ concentrations were usually similar, the different
contributions between three kinds of peroxy radical recombination were caused by different reaction rate
constants. In RACM2, the $HO_2+RO_2$ reaction rate varied from $5.1\times10^{-12}$ $cm^3$ molecule$^{-1}$ s$^{-1}$ (methyl peroxy
radical at 298 K) to $1.6\times10^{-11}$ $cm^3$ molecule$^{-1}$ s$^{-1}$ (isoprene derived $RO_2$ at 298K). In comparison, the effective
$HO_2+HO_2$ reaction rate constant was $3.5\times10^{-12}$ $cm^3$ molecule$^{-1}$ s$^{-1}$ assuming ambient $H_2O$ mixing ratio of 2%.
The self-combination of methyl peroxy radicals rate constant was $3.5\times10^{-13}$ $cm^3$ molecule$^{-1}$ s$^{-1}$, one order of
magnitude smaller than the other radical recombination reaction. The reversible reaction between peroxyacyl
radical and PANs became a net radical sink in the morning because relatively high-$NO_2$ and low-temperature
shifted the thermodynamic equilibrium to form PANs. The net formation of PANs followed by physical
losses contributed on average 12% of $L(RO_x)$. Besides, part of the $RO_2$ species reacts with NO to form
organic nitrate rather than recycle to $HO_2$ radical, resulting in 6% of the radical losses during the daytime.
As for the nighttime, since the radicals formed from $NO_3$ oxidation were dominantly OLND (peroxy radicals
of $NO_3$-alkene adduct reacting via deposition) and OLNN (peroxy radicals of $NO_3$-alkene adduct reacting to
form carbonitrates and $HO_2$) in RACM2, the nighttime radical losses were dominated by the formation of
organic nitrates from OLND and OLNN reaction with themselves and other peroxy radicals. The radical
termination processes in winter were quite different from that in summer. During wintertime, the peroxy
radical recombination was almost negligible, and the radical termination was almost all contributed by the
reactions with NOx (Zhang et al., 2022;Tan et al., 2018d;Ma et al., 2019;Slater et al., 2020).

## 4.2 OH and $HO_2$ measurement-model comparison

OH and $HO_2$ radical concentrations were simulated by a box model, which showed generally good
agreements with observations (Fig. 3). A significant discrepancy between observed and modelled $HO_2$
concentrations occurred on 12 and 13 June. On these two days, maximum $HO_2$ increased to $2.6\times10^9$ cm$^{-3}$,
twice of the campaign averaged maximum, while modelled $HO_2$ concentration remained nearly the same as
the campaign averaged maximum. We investigated the discrepancy between observed and modelled $HO_2$
against different chemical compositions but could not identify the cause of elevated $HO_2$ concentration on
these two days. In the following analysis, the observation-model comparison mainly focused on the mean
diurnal average to extract the overall feature of the campaign.

### 4.2.1 OH underestimation in low NO regime

As shown in Fig. 4, the modelled OH concentration captured the increasing trend in the morning but
unpredicted the measurement since 10:00 with largest discrepancy occurred at noon. The $HO_2$ measurement-
model comparison showed similar diurnal variation but the largest discrepancy shifted to 1 hour later
together with the diurnal maximum. On daytime basis, the modelled OH and $HO_2$ radical concentrations
were on average 30% and 28% smaller than measurements, respectively. The discrepancies can be explained
by their respective combined $1\sigma$ uncertainties of measurement and model calculation (10% and 13% for
measurement and 32% and 40% for model calculation). In fact, the $HO_2$ discrepancy in the mean diurnal
profile was mainly caused by two outlier days, which disappeared in the median diurnal profile (Fig. S6).
However, the discrepancy of OH was also observed in median diurnal profile indicating a persistent OH
underestimation during afternoon.
The OH underestimation discrepancy showed dependence on the NO concentration. Figure 9 illustrates the
dependence of observed and modelled $HO_x$ radicals on NO concentration. To remove the influence of
photolysis on OH radical, OH concentration was normalized to $j(O^1D)$ prior to NO dependence analysis.
The observed median $OH_{norm}$ was almost constant over the whole NO regime, while the modelled value
tended to decrease towards lower NO (<0.3 ppb). The modelled $OH_{norm}$ was 42% smaller than the observed
one at NO mixing ratio below 0.1 ppb (Fig. 9), which was beyond the measurement-model combined
uncertainty. This discrepancy was mainly caused by the data obtained in the afternoon. The observed and
modelled $HO_2$ agreed throughout the NO regime (Fig. 9), and was consistent with the median diurnal profiles.
Such OH-underestimation in low NO regime (typically with NO concentration less than 1 ppb) was
frequently found in environments with intense biogenic emission, especially isoprene (Tan et al., 2001;Ren
et al., 2008;Lelieveld et al., 2008;Whalley et al., 2011;Stone et al., 2011a;Lu et al., 2012;Hofzumahaus et al.,
2009;Lu et al., 2013). We included up-to-date chemical mechanisms related to H-shift processes to consider
the impact of additional OH source, such as the H-shift mechanism of isoprene derived peroxy radicals
(Peeters et al., 2014). However, during this campaign, isoprene concentration was only 0.2 ppb, contributing
5% of the modelled OH reactivity. The H-shift mechanism of isoprene derived peroxy radicals only increased
1.2% of the modelled OH concentration and thus play a minor role in OH chemistry. Therefore, other
processes should account for the OH underestimation in low NO conditions.
To resolve the OH underestimation, a genetic mechanism $X$ was proposed for the Backgarden 2006 campaign,
in which $X$ served as NO that converted $RO_2$ to $HO_2$ and then $HO_2$ to OH (Hofzumahaus et al., 2009).
Sensitivity tests demonstrated the requested amount of $X$ was equivalent to 100 ppt NO for the EXPLORE-
YRD campaign (Fig. 9). Comparatively, the $X$ concentration is the same as in Wangdu campaign (Tan et al.,
2017) but smaller than those identified in Backgarden (0.8 ppb (Hofzumahaus et al., 2009)) , Yufa (0.4 ppb
(Lu et al., 2013)), and Heshan (0.4 ppb (Tan et al., 2019a)), where the biogenic isoprene and OH reactivities
were three to five times and twice as high as during this campaign, respectively (Table 3).
It should be pointed out that the precedingly quantified $X$ of 100 ppt equivalent NO was supposed to be the
lowest limit in this study, if missing reactivity existed. Therefore, we performed a series of sensitivity tests,
by adding a genetic reaction converting OH to $RO_2$ that equivalent to 30% of the total OH reactivity was
added to account for the possible missing reactivity in this study. The adopted degree of missing reactivity
was comparable to that was observed in central Beijing (Whalley et al., 2021), which represented a
significant portion of potential missing reactivity. Besides, the formed $RO_2$ species was varied to investigate
the influence of different $RO_2$ types on the modelled radical concentrations including the $MO_2$ (methyl
peroxy radical), ETEP (peroxy radical formed from ethene), and $ACO_3$ (acetyl peroxy radical). In these cases,
the modelled OH decreased by $1.1\sim1.7\times10^6$ $cm^{-3}$ compared to the base case, and the requested amount of $X$
increased to be equivalent to 200~300 ppt of NO depending on the specific $RO_2$ types (Fig. S7).
On the other hand, the OH measurement-model discrepancy could be attributed to measurement artifacts
(Mao et al., 2012;Novelli et al., 2014;Novelli et al., 2017;Rickly and Stevens, 2018;Fittschen et al., 2019).
Previous studies proposed that stabilized Criegee intermediates (SCIs) produced from reaction of ozone with
alkenes and trioxides (ROOOH) produced from reaction of larger $RO_2$ with OH might cause artificial OH
signals using LIF techniques (Novelli et al., 2017;Fittschen et al., 2019). However, chemical modulation
tests on an ozone polluted day when both $O_3$ and ROOOH (modelled) concentrations were high (7 June)
indicated insignificant interference for OH measurement in this study (Fig. S8). Furthermore, little relevance
of ROOOH and the degree of disagreement between measurement and model was found in this study (Fig.
S9), and thus, there is no hint for significant OH measurement interference during the EXPLORE-YRD
campaign. However, one should note that the precision is not good enough to rule out the possibility.

### 4.2.2 Monoterpenes influence

The observed monoterpenes varied from 0.2 to 0.4 ppb showing a broad peak around noon (Fig. 2). The high
monoterpene concentration and daytime peak indicate a strong daytime source given its short lifetime due
to oxidation (24 minutes for $\alpha$-pinene or 8.2 minutes for Limonene, OH=$1.0\times10^7$ cm$^{-3}$, O$_3$=80 ppb). The
diurnal variation was different from forest environments where maxima usually appeared at night (Kim et
al., 2013;Wolfe et al., 2014;Hens et al., 2014). The relatively low nighttime monoterpenes could be related
the strong NO$_3$ chemistry in this study (Wang et al., 2020a).
In the base model run, observed monoterpenes concentrations were all allocated to $\alpha$-pinene accounting for
0.5 s$^{-1}$ of $k_{OH}$ (Fig. 7). Detailed mechanism referred to $\alpha$-pinene oxidation in RACM2 were listed in Table
S1. A sensitivity test without monoterpenes constrained showed the $k_{OH}$ would decrease by 1.0 s$^{-1}$. Apart
from the decrease in monoterpene itself, half of the decrease of $k_{OH}$ was attributed to the degradation products
of $\alpha$-pinene oxidation. Consequently, the daytime OH and HO$_2$ concentrations would increase by 7% ($5\times10^5$
cm$^{-3}$) and 4% ($3\times10^7$ cm$^{-3}$), respectively (Fig. 4).
We also performed a sensitivity test to attribute the sum of monoterpenes to Limonene, another monoterpene
species in RACM2. In this case, the OH concentration would decrease by 11%, while the HO$_2$ concentration
would slightly increase by 1% relative to the base case. The reduced modelled OH concentration was resulted
from the three times faster reaction rate constant of Limonene with OH ($1.6\times10^{-10}$ cm$^{-3}$ s$^{-1}$ at 298K) than that
of $\alpha$-pinene ($5.3\times10^{-11}$ cm$^{-3}$ s$^{-1}$ at 298K). It indicated that the different assumptions of monoterpenes
speciation had a minor impact on modelled OH and HO$_2$ concentrations in this study.
In recent studies, Whalley et al. (2021) highlighted that large RO$_2$ species, such as those derived from $\alpha$-
pinene and ozone reaction, form RO species upon reaction with NO, and these RO species can isomerize to
form another RO$_2$ species rather than forming HO$_2$ directly, and thus might have impact on the modelled OH
and HO$_2$ concentration. We also performed a sensitivity test to substitute the reactions of $\alpha$-pinene with ozone
in RACM2 by those considering RO isomerization in MCM3.3.1. The modelled OH and HO$_2$ concentrations
decreased by $2.0\times10^4$ cm$^{-3}$ and $2.5\times10^7$ cm$^{-3}$, respectively compared to the base model (Fig. S3), indicating
that $\alpha$-pinene derived RO isomerization had little impact on the modelled OH and HO$_2$ concentrations in this
study.
Other studies conducted in forested environments with a strong influence of monoterpenes from pine trees
emission found discrepancies of up to three times in HO$_2$ measurement-model comparison (Kim et al.,
2013;Wolfe et al., 2014;Hens et al., 2014). In present study, however, HO$_2$ concentration was well
reproduced by chemical model within combined uncertainty during daytime with high monoterpenes
concentrations. Nevertheless, we cannot draw solid conclusion that the monoterpenes oxidation chemistry
in environment with both strong anthropogenic and biogenic influences can be captured by the applied
chemical mechanisms with respect to HO$_x$ concentration, since missing HO$_2$ sources and sinks might exist
simultaneously but cancel out each other. Given that there were no OH reactivity or RO$_2$ observations in this
study, we cannot rule out these possibilities.

### 499    4.2.3 HO$_2$ heterogeneous uptake

A recent model study proposed that HO$_2$ heterogeneous uptake processes play an important role in HO$_x$
radical chemistry and thus suppress ozone formation in China (Li et al., 2019). The RACM2-LIM1
mechanisms used in our study only consist gas phase reactions without heterogeneous chemistry. Therefore,
in this section, we performed a sensitivity test with HO$_2$ radical uptake considered to investigate the potential
impact on the modelled radical concentrations by adding a radical termination process (R6).
HO$_2$ + Aerosol → products                    (R6)
The heterogeneous loss rate of HO$_2$ radical is limited by the free molecular collision because the aerosol
surface is mainly contributed by submicron particles. HO$_2$ radical uptake process can be simplified as a
pseudo first order reaction, and the first-order kinetics constant can be calculated by the Eq. 3:
$k_{HO_2} = \frac{V_{HO_2} \times S_a \times \gamma}{4}$                    (Eq. 3)
$V_{HO_2} = \sqrt{\frac{8RT}{\pi \times 0.033}}$                    (Eq. 4)
$V_{HO2}$ represents the mean molecular velocity of HO$_2$ determined by Eq. 4. $S_a$ is the humid aerosol surface
areas calculated by the SMPS measured particle number and size distribution in each size bin corrected by
the hygroscopic growth factor. $\gamma$ is the effective $HO_2$ uptake coefficient on aerosol giving the probability of
$HO_2$ loss by impacting the aerosol surface.
The effective uptake coefficients vary from $10^{-5}$ to 0.82 from multiple laboratory studies (Thornton et al.,
2008;Taketani et al., 2009;Taketani and Kanaya, 2010;George et al., 2013;Lakey et al., 2015;Zou et al.,
2019). A relatively high value of 0.2 was found in aerosol samples collected in North China Plain, which
was attributed to the abundant dissolved copper ions in aqueous aerosol (Taketani et al., 2012). A study based
on radical experimental budget analysis determined the effective $HO_2$ uptake coefficient to be 0.08±0.13 in
North China Plain (Tan et al., 2020). In our sensitivity tests, both coefficients were applied and simulated
separately.
As shown in Fig. 4, the incorporation of $HO_2$ heterogeneous uptake process worsened the model-
measurement agreement with both OH and $HO_2$ radicals for both cases. The modelled OH and $HO_2$ radicals
were reduced by 10% and 20%, respectively, for the coefficient of 0.2, and by 5% and 10% for the coefficient
of 0.08. For the case the coefficient of 0.08, the increased radical loss rate from $HO_2$ uptake process was 0.4
ppb $h^{-1}$ on daytime basis, which was smaller than that during the Wangdu campaign (0.6±1.3 ppb $h^{-1}$). The
discrepancy between two studies was caused by the lower aerosol surface areas during the EXPLORE-YRD
campaign (750 compared to 1600 $\mu m^2$ $cm^{-3}$). The measured and modelled $HO_2$ concentrations agreed within
33% on daytime basis, which was less than the 40% uncertainty of $HO_2$ simulation. However, this
discrepancy enlarged to 51% as the coefficient increased to 0.2 exceeding the uncertainty of $HO_2$ simulation.
The agreements between measurement and model calculation of OH and $HO_2$ indicated that the base model
without heterogenous reaction captured the key processes for OH and $HO_2$ radical chemistry in this study.
As discussed in Sect. 4.2.1, a series of sensitivity tests had been performed to test the effect of missing
reactivity on the modelled radical concentrations (Fig. S7). It turned out that when OH converted to $MO_2$,
the modelled $HO_2$ would increase by $6.2\times10^7$ $cm^{-3}$ compared to the base case which makes more room for
the $HO_2$ heterogeneous loss. However, considering the potential effect of missing reactivity on $HO_2$, the
measured and modelled $HO_2$ discrepancy (41%) would still be beyond the uncertainty of $HO_2$ simulation for
coefficient of 0.2. On the contrary, for cases that OH converted to ETEP and $ACO_3$, the modelled $HO_2$
decreased by $1.3\times10^7$ $cm^{-3}$ and $1.5\times10^7$ $cm^{-3}$, respectively compared to the base cases, possibly due to the
faster radical termination rates through $RO_2+HO_2$ in both these cases compared to that of $MO_2$. Nevertheless,
the model sensitivity tests suggested that $HO_2$ uptake coefficient was less than 0.2, if the $HO_2$ heterogeneous
loss played a role during this campaign.

## 4.3 local Ozone production rate

Peroxy radical chemistry is intimately tied to the atmospheric ozone production. All peroxy radicals which
could react with NO to form $NO_2$ leading to ozone formation ($F(O_x)$), as expressed in Eq. 5. In this study,
the ozone formation contributing from $RO_2$ was derived from model calculation due to the absence of $RO_2$
measurement. The reaction rate constant between $HO_2$ and NO is approximately $8.5\times10^{-12}$ $cm^3$ molecule$^{-1}$ s$^{-}$
$^{1}$ at 298 K, while the rate constant for the reaction of $RO_2$ with NO varies significantly (ranging in fivefold)
depends on the specific speciation in RACM2. Besides, the $NO_2$ yield from $RO_2$ and NO reaction also differs
for different $RO_2$ groups in RACM2. Part of the $RO_2$ radicals reacts with NO forming organic nitrates rather
than producing $NO_2$ and recycling the peroxy radicals. The nitrate yield increases with higher carbon
numbers and branch structure. Therefore, the $NO_2$ production from $RO_2+NO$ reaction is manipulated by the
effective reaction rate considering both reaction rate constant and $NO_2$ yield for different $RO_2$ species $i$ (Eq.

554    5).

$F(O_x) = k_{HO_2+NO}$ [$HO_2$] [NO] $+ \sum_i k_{RO_2i+NO}$ [$RO_2$]$_i$ [NO]            (Eq. 5)
On the other hand, formed $O_3$ could be involved and consumed in the radical chain reactions by initiating
the radicals from photolysis and reaction with alkenes and propagating the radicals from reaction with OH
and $HO_2$, and besides, part of the $NO_2$ would react with OH to generate nitric acid rather than photolysis
($L(O_x)$). Additionally, $NO_2$ could also react with $O_3$ to form $NO_3$ radical, which could further combine with
another $NO_2$ to form $N_2O_5$ or oxidize VOCs to form organic nitrates, leading to 2 to 3 times faster $O_x$ loss
than $NO_3$ radical formation. Considering the fact that $NO_3$ radical could be easily photolyzed to regenerate
$NO_2$ and $O_3$ or be titrated by NO to regenerate $NO_2$, the contribution from net $NO_3$ radical formation pathway
was taken into account by taking the largest $O_x$ loss per $NO_3$ net formation of 3 in Eq. 6.
$L(O_x) = J(O^1D)$ [$O_3$] $\times \varphi + k_{O_3+Alkenes}$ [Alkenes] [$O_3$] $+ k_{O_3+OH}$ [OH] [$O_3$] $+ k_{O_3+HO_2}$ [$HO_2$] [$O_3$] $+$
$k_{OH+NO_2}$ [OH] [$NO_2$] $+ 3 \times (k_{NO_2+O_3}$ [$NO_2$] [$O_3$] $- k_{NO+NO_3}$ [NO] [$NO_3$] $- j_{NO_3}$ [$NO_3$])     (Eq. 6)
Thus, the net ozone production rate ($P(O_x)$) could be deduced from the difference between $O_x$ formation and
$O_x$ loss rates as expressed in Eq. 7.
$P(O_x) = F(O_x) - L(O_x)$                                                    (Eq. 7)
Figure 10a shows the mean diurnal profiles of the calculated $F(O_x)$ and $L(O_x)$ in this study. Fast ozone
formation rate of up to 20 ppb $h^{-1}$ was observed at 09:00, while the maximum ozone loss rate of 4 ppb $h^{-1}$
shifted to two hours later at noon, when the ozone formation rate reduced to 11.4 ppb $h^{-1}$. This rate was
comparable to other campaigns conducted in rural areas, while the ozone production rates increased
significantly in urban areas, where the noontime ozone formation rates varied from 13.9 ppb $h^{-1}$ in Tokyo to
65 ppb $h^{-1}$ in Mexico (Table 3).
Fast ozone formation is the consequence of both strong primary source and efficient radical propagation.
The latter one can be evaluated by the ratio between $F(O_x)$ and $P(RO_x)$ and known as ozone production
efficiency (OPE). As discussed in Sect. 4.1, the radical primary source was relatively high during the
EXPLORE-YRD campaign, and thus, the OPE was only 1.7, which was smaller than or comparable to other
rural campaigns (Table 3). Urban campaigns in the U.S., Mexico and Tokyo showed significant higher OPE
varying from 6 to 10 (Table 3) probably benefit from the moderate $NO_x$ level. In comparison, OPE was
smaller in four megacities in China (Beijing: 3.4, Shanghai: 3.1, Guangzhou: 2.2, Chongqing: 3.6) than in
the U.S. cities ranging from 3 to 7 because of the suppression of high $NO_x$ in Chinese cities (Tan et al.,
2019b). However, during the EXPLORE-YRD campaign, the low OPE indicates that the radical propagation
chain length was relatively short due to low NO conditions.
As shown in Fig. 10b, the integrated net ozone production was 68.3 ppb $d^{-1}$ over the entire daytime (08:00-
16:00). The daily integrated $P(O_x)$ calculated based on the modelled peroxy radicals was 6.9 ppb lower than
on derived from observation (Fig. 10b). The discrepancy for observation and model derived $P(O_x)$ mainly
appears at NO concentration larger than 1 ppb (Fig. 9). This behavior has been observed in a number of
previous urban radical measurement campaigns (Kanaya et al., 2008;Kanaya et al., 2012;Martinez, 2003;Ren
et al., 2003a;Ren et al., 2013;Elshorbany et al., 2012;Brune et al., 2016;Whalley et al., 2018;Tan et al., 2017),
which was caused by the model underprediction of the observed $HO_2$ concentrations under high NO
concentration (typically NO greater than 1 ppb). Although some of the previous $HO_2$ measurement might
suffer from unrecognized interference from $RO_2$ species, this kind of interference have been minimized by
lowering down the added NO concentration in recent studies (Griffith et al., 2016;Brune et al., 2016).
However, the underestimation of ozone production from $HO_2$ radical persist, indicating that the
photochemical production mechanism of ozone under polluted urban environment is still not well understood.
We also investigated the impact of different model scenarios on $P(O_x)$ by comparing integrated $P(O_x)$ in
different cases to that obtained in base model (Fig. 10b). Sensitivity test without $\alpha$-pinene constrained
predicted 6.3 ppb less daily integrated net ozone production than base case. Meanwhile, the contribution of
$\alpha$-pinene derived peroxy radicals (APIP) on $F(O_x)$ only accounted for 2.3 ppb $O_3$ formation (Fig. 10a). The
difference can be attributed to the degradation products of $\alpha$-pinene which also contribute to ozone
production. For example, aldehyde (ALD) is an important daughter product from $\alpha$-pinene oxidation, which
reacts with OH and forms acyl peroxy radicals. Acyl peroxy radicals have two advantages in ozone formation.
On one hand, acyl peroxy radicals have the fastest rate constants with NO among all the peroxy radicals
(2~5 times faster than others). On the other hand, acyl peroxy radicals react with NO to produce $NO_2$ and
methyl or ethyl peroxy radicals, which can further oxidize the NO to $NO_2$ and generate $HO_2$. Given that the
modelled $HO_2$ concentration increased by 4% in the sensitivity test, the smaller in $P(O_x)$ was mainly
attributed to significant reduction in modelled $RO_2$ concentration. In fact, the modelled $RO_2$ concentration
would reduce by 23% if $\alpha$-pinene was not constrained to observation, which indicated $\alpha$-pinene was an
important $RO_2$ precursor. It proved that monoterpene contributes significantly to the photochemical
production of $O_3$ in this study.
Moreover, we also investigated the impact of the $\alpha$-pinene derived RO species which can isomerize to form
another $RO_2$ rather than forming $HO_2$ directly on the calculated ozone production rate. It turned out that
including $\alpha$-pinene derived RO isomerization mechanism in the model run would reduce the daily net $O_3$
production by 1 ppb.
Additionally, $HO_2$ heterogeneous uptake process in the model run would reduce the daily net $O_3$ production
by 4.8 ppb by assuming the effective coefficient of 0.08. The reduction in $P(O_x)$ was only slightly smaller
than the relative change in modelled $HO_2$ concentration (10%) because 62% of the $F(O_x)$ was contributed by
the reaction of $HO_2$ with NO (Fig. 10a).

## 4 Conclusion

A comprehensive field campaign to elucidate the atmospheric oxidation capacity in Yangtze River Delta in
China was carried out in summer 2018, providing the first OH and $HO_2$ radicals observations in this region.
Daily maximum concentrations of OH and $HO_2$ radicals were in the range from 8 to $24 \times 10^6$ $cm^{-3}$ and 4 to
$28 \times 10^8$ $cm^{-3}$, with mean values of $1.0 \times 10^7$ $cm^{-3}$ and $1.1 \times 10^9$ $cm^{-3}$, respectively. The OH radical was of the
second highest concentration among the observations in China, indicating the strong oxidation capacity in
YRD region from the perspective of OH radical concentration. The modelled $k_{OH}$ varied from 5 $s^{-1}$ to 40 $s^{-1}$
over the whole campaign, and 40% of which could be explained by OVOCs, in which measured and
modelled OVOCs made up comparable contributions.
The radical primary source was dominated by HONO photolysis during this campaign, contributing 42% of
$P(RO_x)$. The secondary contributor was the photolysis of carbonyl compounds (including HCHO),
accounting for 24% of the total radical primary source. Radical termination was dominated by the reactions
with $NO_x$ in the morning and peroxy radical self-reactions in the afternoon. Specifically, $OH+NO_2$ reaction
and peroxy radical self-reaction from $HO_2+RO_2$ were the most important pathways, contributing 25% and
33% of the total radical loss rates, respectively.
The comparison between observation and box model simulation showed generally good agreement for both
OH and $HO_2$ radicals on average. However, the OH radical showed a tendency of underestimation towards
low NO regime (NO< 0.1 ppb), and the discrepancy (42%) was beyond the measurement-model combined
uncertainty. The up-to-date H-shift mechanism of isoprene derived peroxy radicals could not explain the
discrepancy due to the low isoprene concentration (0.2 ppb) during this campaign. A genetic OH recycling
process equivalent to 100 ppt NO was capable to fill the gaps, which was also found in previous campaigns
in Backgarden, Yufa, Heshan, and Wangdu in China. In addition, the good simulation in $HO_2$ radical was
different from other monoterpene-rich forest environments, where $HO_2$ underestimations were found.
Additional sensitivity tests were performed to investigate the impact of monoterpenes and $HO_2$
heterogeneous uptake on radical chemistry in this study. Model simulation without monoterpene input or
allocating monoterpene to a different isomer ($\alpha$-pinene and Limonene in this study) showed that $HO_x$ radical
concentrations were not sensitive to the monoterpene in this study. In fact, the modelled $RO_2$ radical
concentration would be reduced by 23% without monoterpene constrained. The reduced $RO_2$ radical offset
the enhancement of $HO_x$ radicals. The combined influence caused the net daily integrated ozone production
to decrease by 6.3 ppb compared to the base model of 61.4 ppb, which demonstrated the importance of

monoterpene chemistry on the photochemical ozone production in this study. The role of $HO_2$ heterogeneous uptake was tested by adding a pseudo first-order reaction loss of $HO_2$, and taking the effective uptake coefficients of 0.2 and 0.08, respectively. The sensitivity test suggested the applied chemical mechanism without $HO_2$ heterogeneous uptake could capture the key processes for $HO_x$ radicals, and the effective uptake coefficient should be less than 0.2, if the $HO_2$ heterogeneous loss played a role in this study, otherwise, the $HO_2$ measurement-model discrepancy would be beyond the combined uncertainty. The daily integrated net ozone production would reduce by 4.8 ppb, if the effective uptake coefficient was assumed to be 0.08.

Additionally, the noontime ozone production rate was 11.4 ppb $h^{-1}$, which was much slower than other campaigns in urban and suburban areas varying from 13.9 to 65 ppb $h^{-1}$. Thus, the ozone production efficiency calculated from the ratio of $P(O_x)$ and $P(RO_x)$ was only 1.7 in this study, which was comparable to the values in rural campaigns but was 3 to 7 times lower than the values in other urban and suburban campaigns, indicating the slow radical propagation rate and short chain length in this study.

***Data availability.*** The data used in this study are available from the corresponding author upon request (k.lu@pku.edu.cn).

***Author contributions.*** YZ and KL organized the field campaign. KL and YZ designed the experiments. XM and ZT analyzed the data. XM wrote the manuscript with input from ZT. All authors contributed to measurements, discussing results, and commenting on the manuscript.

***Competing interests.*** The authors declare that they have no conflict of interest.

***Acknowledgements.*** We thank the support by the Beijing Municipal Natural Science Foundation for Distinguished Young Scholars (Grants No. JQ19031), the National Research Program for Key Issue in Air Pollution Control (Grants No. 2019YFC0214801, 2017YFC0209402, 2017YFC0210004, 2018YFC0213801), the National Natural Science Foundation of China (Grants No. 21976006, 91544225, 91844301).

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

 **Table 1. Measured species and performance of the instruments.**

| Parameters | Techniques | Time resolutions | Limit of Detection[a] | Accuracy |
|---|---|---|---|---|
| OH | LIF[b] | 30 s | $6.0 \times 10^5$ cm$^{-3}$ | ±10% |
| HO$_2$ | LIF[b,c] | 30 s | $1.0 \times 10^7$ cm$^{-3}$ | ±13% |
| Photolysis frequencies | Spectroradiometer | 9 s | [d] | ±10% |
| O$_3$ | UV photometry | 60 s | 0.5 ppb | ±5% |
| NO | Chemiluminescence | 60 s | 60 ppt | ±20% |
| NO$_2$ | Chemiluminescence[e] | 60 s | 0.3 ppb | ±20% |
| HONO | LOPAP[f] | 60 s | 10 ppt | ±20% |
| CO | Infrared absorption | 60 s | 1 ppb | ±1 ppb |
| SO$_2$ | Pulsed UV fluorescence | 60 s | 0.1 ppb | ±5% |
| VOCs[g] | GC-FID/MS[h] | 1 h | 20-300 ppt | ±15% |
| HCHO | Hantzsch fluorimetry | 60 s | 25 ppt | ±5% |
| Glyoxal | CEAS | 60 s | 60 ppt | ±10% |
| Monoterpene[i] | PTR-MS | 10 s | 20 ppt | ±15% |
| PNSD | SMPS | 5 min | 14 nm-700 nm | ±20% |

[a] Signal-to-noise ratio =1. [b] Laser Induced Fluorescence. [c] Chemical conversion to OH via NO reaction before detection. [d]
Process-specific, 5 orders of magnitude lower than maximum at noon. [e] Photolytic conversion to NO before detection, home-
built converter. [f] Long-path absorption photometry. [g] VOCs including C$_2$-C$_{11}$ alkanes, C$_2$-C$_6$ alkenes, C$_6$-C$_{10}$ aromatics. [h] Gas
chromatography equipped with a mass spectrometer and a flame ionization detector. [i] the sum of monoterpene.



 **Table 2. Assignment of measured and constrained VOCs in RAMC2 during this study.**

| RACM | Measured hydrocarbons |
|------|----------------------|
| ACE | acetylene |
| ETH | ethane |
| HC3 | propane, *i*-butane, *n*-butane, 2,2-dimethylbutane |
| HC5 | *i*-pentane, *n*-pentane, cyclopentane, 2,3-dimethylbutane, 2-methylpentane, 3-methylpentane, MTBE, *n*-hexane, 2,3-dimethylpentane, 2,4-dimethylpentane, methylcyclopentane, 2-methylhexane |
| HC8 | cyclohexane, 3-methylhexane, 2,2,4-trimethylpentane, 2,3,4-trimethylpentane, *n*-heptane, methylcyclohexane, 2-methylheptane, 3-methylheptane, *n*-octane, *n*-nonane, *n*-decane, *n*-undecane |
| ETE | ethylene |
| OLI | *trans*-2-butene, *cis*-2-butene, *trans*-2-pentene, *cis*-2-pentene |
| OLT | propene, 1-butene, 1-pentene, 1-hexene, styrene |
| DIEN | 1,3-butadiene |
| BEN | benzene |
| TOL | toluene, ethylbenzene, *i*-propylbenzene, *n*-propylbenzene |
| XYO | *o*-xylene, *o*-ethyltoluene |
| XYM | *m*-ethyltoluene, 1,3,5-trimethylbenzene, 1,2,4-trimethylbenzene, 1,2,3-trimethylbenzene, *m*-diethylbenzene |
| XYP | *m,p*-xylene, *p*-ethyltoluene, *p*-diethylbenzene |
| ISO | isoprene |
| API | sum of monoterpenes |
| HCHO | formaldehyde |
| ACD | acetaldehyde |
| GLY | glyoxal |
| ACT | acetone |
| MACR | methacrolein |
| MVK | methyl vinyl ketone |
| MEK | methyl ethyl ketone |




**Table 3. Summary of filed measurements and model simulation for $j(O^1D)$, $O_3$, $NO_x$, OH, $HO_2$, $P(RO_x)$, $F(O_x)$ and OPE at local noon in urban and**
**suburban environments.**

| Location | Month Year | Type | $j(O^1D)$ /$10^{-5}$ s$^{-1}$ | $O_3$ /ppb | $NO_x$ /ppb | OH /$10^6$ cm$^{-3}$ | $HO_2$ /$10^8$ cm$^{-3}$ | $P(RO_x)$ /ppb/h | $F(O_x)$ /ppb/h | OPE[s] | References |
|---|---|---|---|---|---|---|---|---|---|---|---|
| Pabstthum, Germany, 52.85°N, 12.94°W, 50 km NW of Berlin | July-August 1998 | Rural | 1.5 | 42 | 1.55 | 3.5 | 2.2 | 1.7[a] | 2.2[b] | 1.3 | (Holland et al., 2003;Volz-Thomas et al., 2003;Platt et al., 2002) |
| Nashville, USA, 36°11.4'N, 86°42.0'W, 8 km NE of downtown area | June-July 1999 | Suburban | 3.0[a] | 60[a] | 4.4[a] | 10 | 7.5 | 1.1 | 9[c] | 8.2 | (Martinez, 2003;Thornton et al., 2002) |
| La Porte, USA, 29°40'N, 95°01'W, 40 km SE of Houston | August-September 2000 | Suburban | 3.0 | 70 | 6 | 20 | 7.5 | 4.9 | 25[d] | 5.1 | (Mao et al., 2010) |
| New York (Queens College), USA, 40°44'15''N, 73°49'18''W, in the Borough of Queens | June-August 2001 | Urban | 2.5 | 48 | 28 | 7.0[e] | 1.0[e] | 4.8 | 34[d] | 7.1 | (Mao et al., 2010;Ren et al., 2003b;Ren et al., 2003a) |
| Mexico City, Mexico, 19°25'N, ~7 km SE of downtown area | April-May 2003 | Urban | 4.5 | 115 | 18 | 12[f] | 15[f] | 8.6 | 65[d] | 7.6 | (Mao et al., 2010;Shirley et al., 2006) |
| Essex (Writtle College), England, 51°44'12''N, 0°25'28''E, 25 miles NE of central London | July-August 2003 | Rural | 1.0[g] | 46.5[g] | 10.8[g] | 2[g] | 0.7[g] | 11.6[g] | 7.2[g,h] | 0.6 | (Emmerson et al., 2007) |

| Location | Date | Type | | | | | | | | | Reference |
|---|---|---|---|---|---|---|---|---|---|---|---|
| Tokyo (University of Tokyo), Japan, 35°39'N, 139°41'E, near city center | July-August 2004 | Urban | 2.5 | 32 | 12 | $6.3^e$ | $1.4^e$ | 2.2 $(6.8)^i$ | $13.9^i$ | 6.3 $(2.0)^i$ | (Kanaya et al., 2007;Kanaya et al., 2008) |
| Backgarden, China, 23.487°N, 113.034°E, 60 km NW of downtown Guangzhou | July 2006 | Rural | 3.5 | 51 | 11.4 | 14 | $17^k$ | 10.7 | $18^l$ | 1.7 | (Lu et al., 2012;Lou et al., 2010) |
| Yufa, China, 39.5145°N, 116.3055°E, ~40 km south of the Beijing downtown area | August-September 2006 | Rural | 1.8 | 71 | 8.8 | 5.5 | $7.2^k$ | 7.0 | $15^l$ | 2.1 | (Lu et al., 2013) |
| Mexico City, Mexico, 19°N, 100°W, ~7 km SE of downtown area | March 2006 | Urban | 4.0 | 90 | 49 | $4.6^e$ | $1.9^e$ | 7.5 | $31^c$ | 4.1 | (Dusanter et al., 2009a;Dusanter et al., 2009b;Molina et al., 2010) |
| University of Houston (70 m above ground level), USA, 29.7176°N, 95.3413°W, 5 km SE of downtown Houston | August-September 2006 | Urban (Tower) | 3.1 | 68 | 4 | 15 | 12.5 | 5.3 | $45^d$ | 8.5 | (Mao et al., 2010) |
| University of Houston (70 m above ground level), USA, 29.7176°N, 95.3413°W, 5 km SE of downtown Houston | April-May 2009 | Urban (Tower) | - | 47 | 2.5 | $8.8^e$ | $6.3^e$ | 3 | $18^j$ | 6 | (Ren et al., 2013;Lee et al., 2013) |
| Paris, France, 48.718°N, 2.207°E, ~14 km SW of Paris | July 2009 | Suburban | 2.2 | 35 | 4.3 | 4.2 | $1.3^m$ | $0.75^n$ | $7.1^o$ | 9.5 | (Michoud et al., 2012) |

| Location | Date | Type | | | | | | | | | Reference |
|---|---|---|---|---|---|---|---|---|---|---|---|
| Pasadena, USA, 34.1408°N, 118.1223°W, ~18 km NE of downtown | May-June 2010 | Suburban | 2.1 (2.5)[p] | 45 (72)[p] | 19 (9)[p] | 3.5 (4.0)[p] | 2.0 (5.0)[p] | 4.0 (5.3)[p] | 33 (23)[p,q] | 8.3 (4.3) | (Griffith et al., 2016) |
| London, England, 51°31′16″N, 0°1248″W, in central London | July-August 2012 | Urban | - | 24.2 (37.4)[r] | 13.1 (24.3)[r] | 2.1 (3.0)[r] | 2.0 (0.6)[r] | 4.9 | 5.6[g] | 1.1 | (Whalley et al., 2018;Whalley, 2016) |
| Wangdu, China, 38.71°N, 115.15°E, ~35 km SW of Baoding and 170 km SW of Beijing | June-July 2014 | Rural | 1.8 | 88 | 8.2 | 8.3 | 7.7 | 4.8 | 14.7[b] | 3.1 | (Tan et al., 2017) |
| Heshan, China, 22.728°N, 112.929°E, ~6 km SW of the city of Heshan and 50 km SW of Guangzhou and Foshan | October-November 2014 | Suburban | 1.3 | 51 | 26.9 | 4.8 | 2.3 | 5.1 | 18.1[b] | 3.5 | (Tan et al., 2019a) |
| Beijing, China, 39.97°N, 116.38°E, in central Beijing | May-June 2017 | Urban | 2.4 | 100 | 25 | 9.0 | 3.0 | 7.0 | 7.8[t] | 2.4[t] | (Whalley et al., 2021;Shi et al.,2019) |
| Taizhou, China, 32.56°N, 119.99°E, ~200 km NW of Shanghai | May-June 2018 | Suburban | 2.1 | 82 | 3.6 | 10.6 | 11.4 | 6.8 | 11.4[j] | 1.7 | This study |

[a] Take from a typical day. [b] Calculated from measured $HO_2$ with NO. [c] Calculated from measured peroxy radical with NO reaction. [d] Calculated from measured $HO_2$ and scaled $RO_2$ (measured $HO_2$ times the ratio of modelled $RO_2$ to $HO_2$) with NO. [e] Median. [f] Median and revised. [g] 11:00-15:00 mean. [h] Calculated by summing all of the reaction rates for NO to $NO_2$ conversions. [i] For smog-free day and smog day (in parenthesis) separately. [j] Calculated from measured $HO_2$ and modelled $RO_2$ with NO. [k] $HO_2^*$ ($HO_2$ and partial $RO_2$). [l] Calculated from modelled $HO_2$ and $RO_2$ with NO. [m] Total peroxy radicals ($HO_2+RO_2$). [n] 8:00-16:00 mean. [o] Calculated by measured total peroxy radicals ($HO_2+RO_2$) with NO. [p] For week days and weekend days (in parenthesis) separately. [q] Calculated from measured $HO_2^*$ with NO. [r] For westerly flow and easterly flow (in parenthesis) separately. [s] Calculated by the ratio between $F(O_x)$ and $P(RO_x)$. [t] Daily mean.

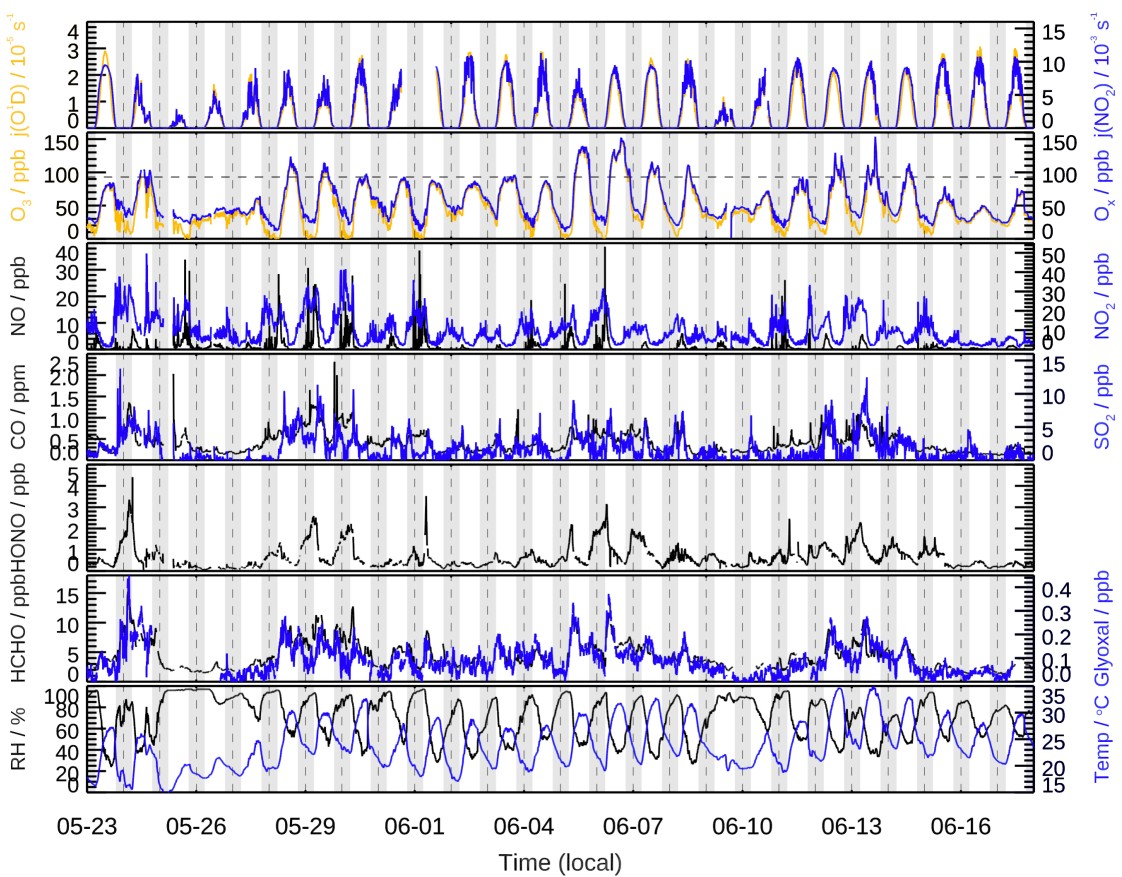



**Figure 1. Time series of measured photolysis frequencies (j(O¹D), j(NO₂)), relative humidity (RH),**
**ambient temperature (T), and concentrations of O₃, Oₓ (=O₃+NO₂), NO, NO₂, CO, SO₂, HONO,**
**formaldehyde (HCHO), and glyoxal (CHOCHO). The dotted horizontal line represents the Chinese**
**national air quality standard level II of O₃ (hourly averaged limit 93 ppb). The grey areas denote**
**nighttime.**

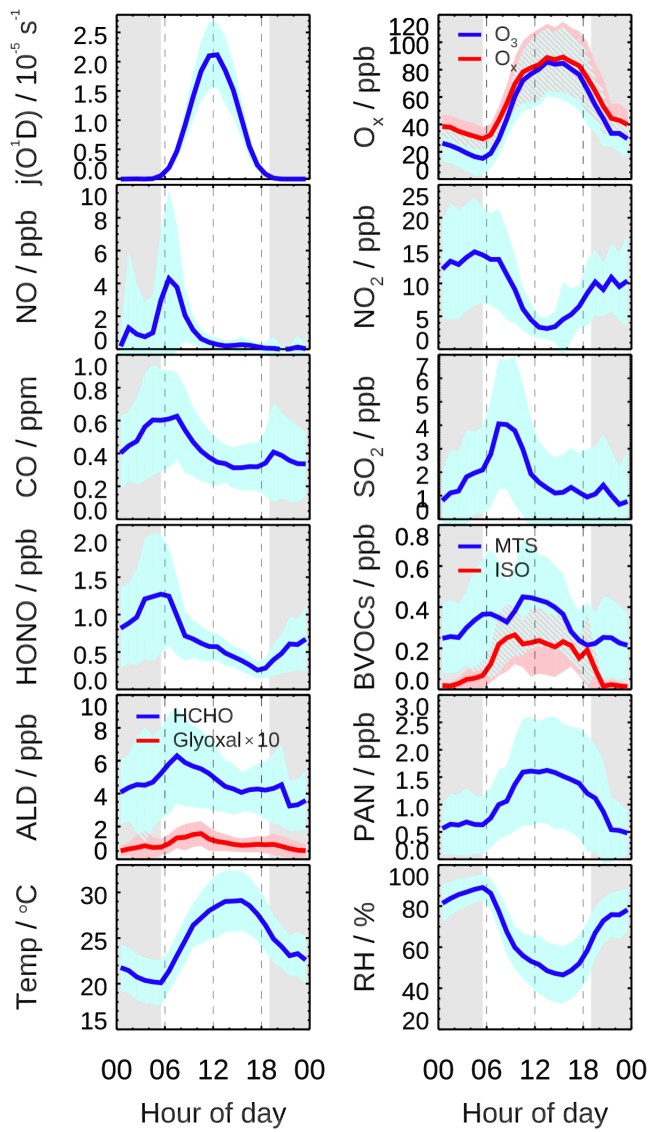


**Figure 2. Mean diurnal profiles of measured photolysis frequencies ($j(O^1D)$), relative humidity (RH),**
**ambient temperature (T), and concentrations of $O_3$, $O_x$ (=$O_3$+$NO_2$), NO, $NO_2$, CO, $SO_2$, HONO,**
**formaldehyde (HCHO), glyoxal (CHOCHO), biogenic VOCs (monoterpenes, isoprene), and PAN.**
**Data are averaged over the period with $HO_x$ radical measurement. Colored areas denote the standard**
**deviation of variability (1σ). The grey areas denote nighttime.**






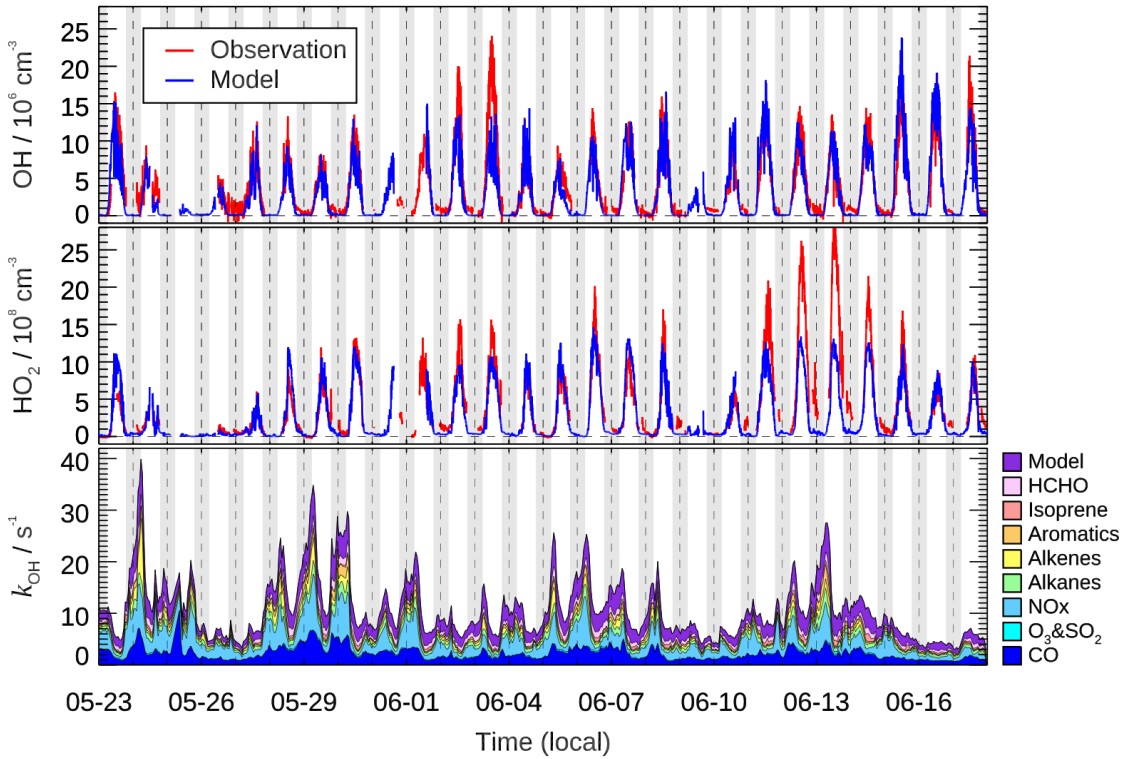


**Figure 3. Time series of observed and modelled OH and HO₂ concentrations, and the modelled grouped OH reactivity ($k_{OH}$). Vertical dash lines denote midnight. The grey areas denote nighttime.**


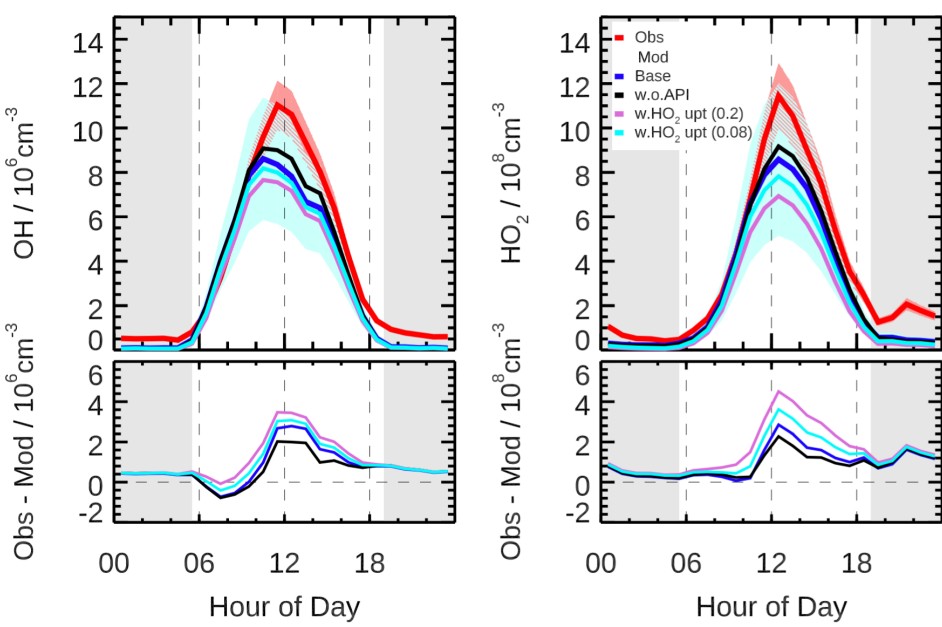


**Figure 4. The mean diurnal profiles of measured and modelled OH and HO₂ concentrations (upper panel) as well as the discrepancies between observation and model (lower panel) in different scenarios (Scenario1: base case; Scenario2: without α-pinene constrained; Scenario 3: with HO₂ heterogeneous uptake process considered by assuming the uptake coefficient of 0.2; Scenario 4: with HO₂ heterogeneous uptake process considered by assuming the uptake coefficient of 0.08). Colored areas denote 1σ uncertainties of measured (red) and base case modelled (blue) radical concentrations, respectively. The grey areas denote nighttime.**

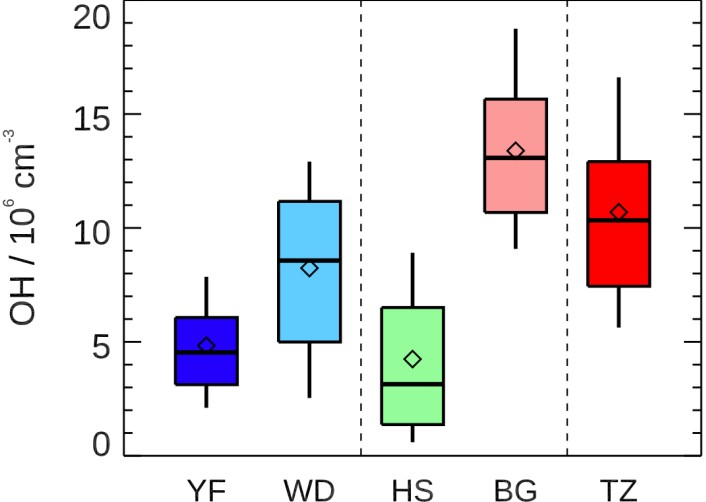

**Figure 5. Summary of OH radical concentrations (noon time, 11:00-13:00) measured in five summer field campaigns in China. Yufa (YF) and Wangdu (WD) campaign in North China Plain, Heshan (HS) and Backgarden (BG) campaign in Pearl River Delta, and Taizhou (TZ, this study) campaign in Yangtze River Delta. The box-whisker plot shows the 90th, 75th, 50th, 25th, and 10th percentile values of noon OH radical concentrations in each campaign. The diamond shows the mean values of noon OH radical concentrations.**

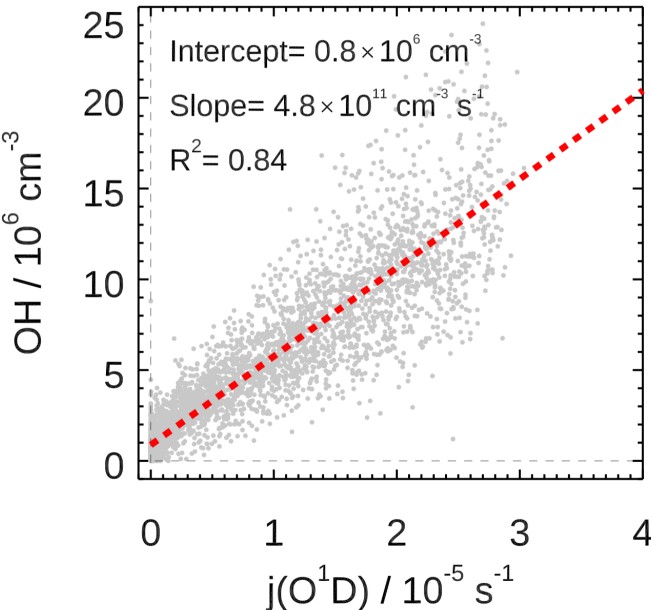

1132

**Figure 6. Correlation between measured OH and j(O¹D). Grey scatter plot represents the 5 min**
**observation result for the EXPLORE-YRD campaign. A linear fit which takes both measurements**
**error into account is applied. The linear fit lines and correlation slopes, intercept and coefficients are**
**also shown.**

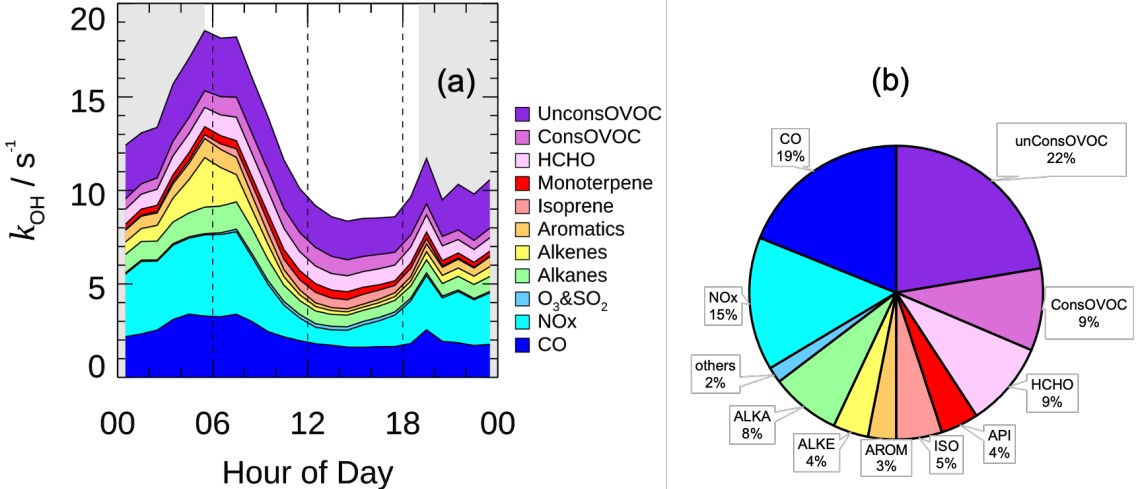


**Figure 7. (a) The mean diurnal profiles of speciated OH reactivity. The grey areas denote nighttime.**
**(b) Breakdown of modelled OH reactivity for daytime conditions (08:00-16:00).**


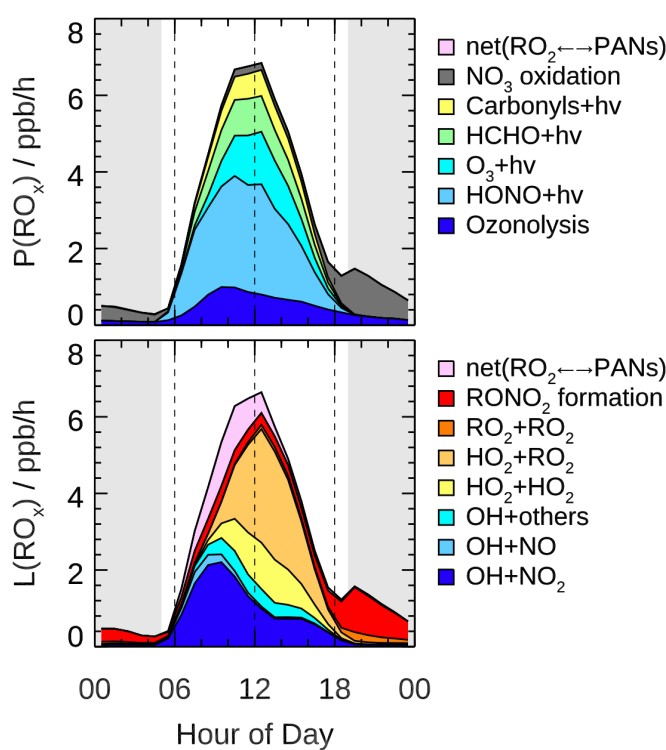


**Figure 8. Hourly mean diurnal profiles of primary sources and sinks of RO$_x$ radicals from model**

**calculations. The grey areas denote nighttime.**





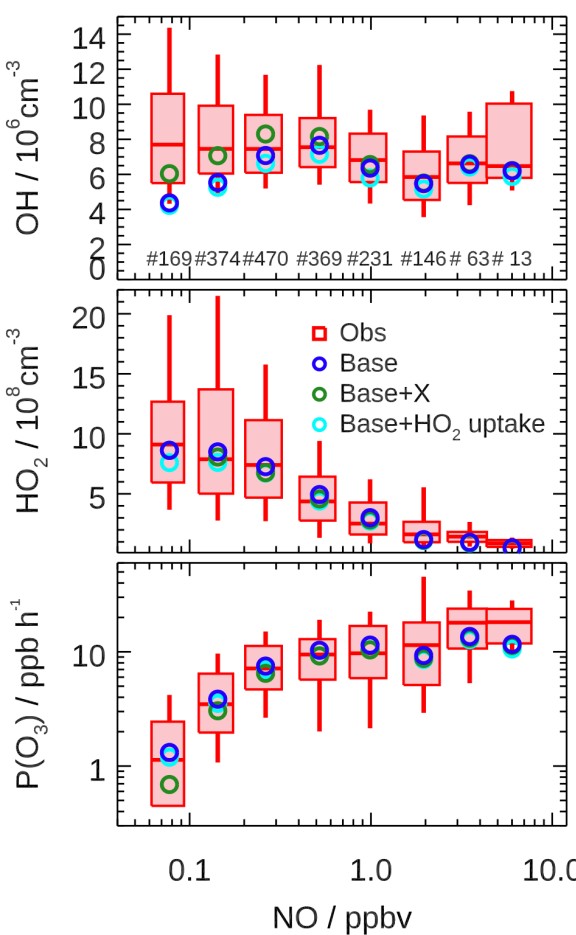


**Figure 9. Dependence of measured and modelled OH, HO₂, and P(Oₓ) on NO concentrations for**

**daytime condition (j(O¹D)> 0.5×10⁻⁵ s⁻¹). Box-whisker plot shows the median, the 75 and 25 percentiles,**

**and the 90 and 10 percentiles of the measured results for each NO interval bins. Only median values**

**are shown for modelled results. Numbers in upper panel represent the data points incorporated in**

**each NO interval. Results from base case, with additional recycling process by a species *X* (equivalent**

**to 100 ppt NO), and with additional HO₂ heterogeneous uptake process (*γ* assuming of 0.08) are all**

**plotted.**

1156

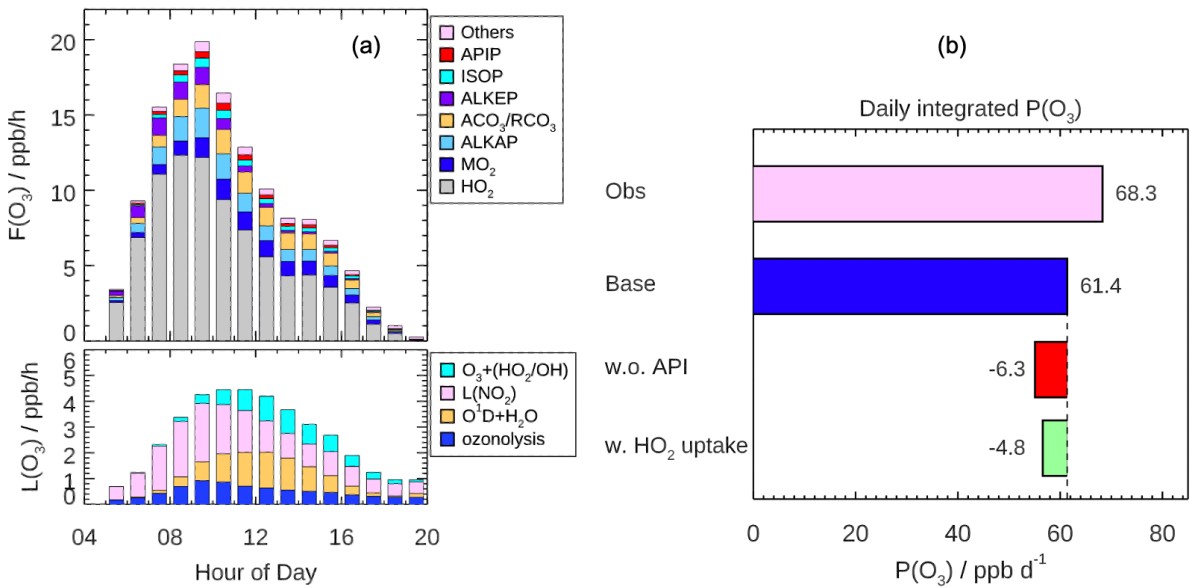

1157

**Figure 10. (a) Mean diurnal profiles of the speciation ozone formation rate ($F(O_x)$) from different peroxy radical species (upper panel) and the speciation ozone destruction rate ($L(O_x)$, lower panel) calculated based on the measured OH and $HO_2$ and modelled $RO_2$ radicals. (b) Daily (08:00-16:00) integrated net ozone production calculated from the observed and modelled radical concentration, respectively. The discrepancies between two model scenarios run (Scenario1: without $\alpha$-pinene constrained; Scenario2: with $HO_2$ heterogeneous uptake considered by assuming $\gamma$ of 0.08) from base case are also shown.**