# Peer review of "OH and HO2 radical chemistry at a suburban site during the EXPLORE-YRD campaign in 2018"

_Atmospheric Chemistry and Physics, 2021_

## Author Comment (AC1)

Dear Christa,

Thanks for your comments! According to your suggestions, we tested the importance of ROOOH on the degree of disagreement between measurement and model in this study by integrating the reactions of OH with all $RO_2$ species except for $CH_3O_2$ with a rate constant of $1.5×10^{-10}$ cm$^3$ s$^{-1}$ in our model, and adding a loss process for ROOOH with a rate of $10^{-4}$ s$^{-1}$, corresponding to a lifetime of around 3 hours as suggested by Fittschen [1,2], to represent an upper limit of ROOOH formation.

As shown in Figure 1, the modelled ROOOH concentration peaked around 14:00 with the peak values on the order of $10^9~10^{10}$ cm$^{-3}$, which were comparable to the modelled values by Fittschen, et al.[2]. However, the correlation of modelled ROOOH with the ratio of observed to modelled OH and the difference between observed and modelled OH showed little relevance ($R^2$=0.01 and 0.03, respectively on daytime basis, Figure 2). It indicated that the ROOOH interference was not able to explain the disagreement between measurement and model in this study. Besides, the modelled ROOOH concentrations were high on the day when the chemical modulation tests were performed (7$^{th}$ June, Figure 1). If ROOOH did cause significant OH interference in our FAGE system, it would also cause significant OH interference on that day, which however, was not discovered in the chemical modulation tests. It was also worth noting that incorporating the additional reactions of $RO_2$+OH into the existed reaction mechanism had little influence on the modelled OH, $HO_2$, and $RO_2$ concentrations. Nevertheless, ROOOH could be an OH interference in FAGE system, but as pointed also by Fittschen, et al. [2], the occurrence of this interference is highly dependent on the design and measurement conditions of different FAGE instruments. This interference was not the reason for the disagreement between model and measurement in our FAGE system at least for this campaign.

Thanks again for your insightful comments, and we will add some discussions and references as you mentioned.

[Figure]

Figure 1. Time series of modeled ROOOH concentrations during EXPLORE-YRD campaign in 2018.

[Figure]

Figure 2. Dependences of the ratio of observed to modeled OH (a) and the difference between observed and modeled OH (b) on the concentration of ROOOH during daytime periods (08:00-16:00)

1    Fittschen, C. The reaction of peroxy radicals with OH radicals. *Chem Phys Lett* **725**, 102-108 (2019).

2    Fittschen, C. *et al.* ROOOH: a missing piece of the puzzle for OH measurements in low-NO environments? *Atmospheric Chemistry and Physics* **19**, 349-362 (2019).

---

## Author Comment (AC2)

**General comments:**

*The paper presents OH and HO$_2$ observations made in the Yangtze River Delta and compares to model predictions using a box model constrained with the RACM2-LIM1 mechanism. The model-measurement comparison highlighted that OH concentrations were under-predicted by the model under low NO conditions, whilst modelled HO$_2$ agreed reasonable well with the HO$_2$ observed. The impact of monoterpenes on radical concentrations and ozone production was investigated as was heterogeneous HO$_2$ loss. Generally, the results were well presented and the manuscript was reasonably easy to follow although the English used could be improved on. The manuscript would be improved considerably if the results presented were discussed in the context of a wider breadth of previous literature and I have tried to highlight a number of papers that are of relevance to this work in my comments below. I recommend publication once the following comments have been addressed.*

**Answer:**

We would like to thank the reviewer for the comments and questions which helped us to improve the manuscript. The reviewer comments are given below together with our responses and changes made to the manuscript. The technical comments were changed accordingly and not listed below for simplification.

**Major Comments:**

1. *Pg 3, lines 73 – 78: There was also a field campaign in Beijing in the summer of 2017 (Whalley et al., ACP, 21, 2125 – 2147, 2021) where OH, HO2, total RO2 and kOH observations were made and compared to box model predictions. This work should be discussed and referenced in the context of previous radical measurement studies conducted in the summer in China, particularly in light of the elevated OH concentrations observed. The Beijing results should also be added to Table 3.*

**Answer:**

We added the HOx observation conducted in Beijing in the summer of 2017 in the context of previous radical measurement studies and updated the Table 3 accordingly.

We revised the sentences as 'Six field campaigns have been implemented in China during summer periods, namely the Backgarden (2006), Heshan (2014), Shenzhen (2018) campaigns in Pearl River Delta (PRD) (Lu et al., 2012;Tan et al., 2019;Wang et al., 2019), and Yufa (2006), Wangdu (2014), and Beijing campaigns in North China Plain (NCP) (Lu et al., 2013;Tan et al., 2017;Whalley et al., 2021)…'

2. *Pg 5, lines 127 – 133: The authors state the conversion efficiency of HO2 to OH at 5 ppm NO to be 20%, but should also state the conversion efficiency at 2.5 ppm NO. From laboratory tests, what is the conversion efficiency of an alkene RO2 to OH at 5 ppm and 2.5 ppm NO? The authors should discuss these details in relation to what is seen in other FAGE instruments, e.g. Fuchs et al., AMT, 4, 1209-1225, 2011 and Whalley et al., AMT, 6, 3425 – 3440, 2013.*

**Answer:**

We added some discussion about the $HO_2$ interference from $RO_2$ radical as the reviewer suggested.

We revised the second paragraph in section 2.2 as 'Previous studies indicated that part of the $RO_2$ species derived from longer chain alkanes (> C3), alkenes, and aromatic compounds have the potential to rapidly convert to OH on the same time scale as $HO_2$ inside the fluorescence cell, and thus, might cause interference for $HO_2$ measurement (Fuchs et al., 2011;Whalley et al., 2013). To minimize the potential interference from $RO_2$, the added maximum NO mixing ratio was chosen to be 5 ppm, resulting in the maximum $HO_2$ conversion efficiency being 20%. Furthermore, the NO injection was switched between 2.5 ppm and 5 ppm every 2 minutes, corresponding to the $HO_2$ conversion efficiencies of 10% and 20%, respectively. If $RO_2$ interference was significant, the $HO_2$ measurement would be different between two NO injection modes. The $HO_2$ measurements with different NO injection rates only showed a difference of 6%, indicating that the potential interference from $RO_2$ was within the $HO_2$ measurement uncertainty (13%) during this campaign.'

3. *Pg 6, lines 140 – 147: Some key details on the chemical removal technique should be added to this paper. The concentration of propane added, the % removal of ambient OH, discussion of any losses of ambient OH to the chemical modulation device and % removal of OH internally. Again, the authors should discuss these details in relation to what is seen in other FAGE instruments, e.g. Woodward-Massey et al., AMT, 13, 3119 – 3146, 2020 and Cho et al., AMT, 14, 1851 – 1877, 2021.*

**Answer:**

We added some details on the chemical removal technique as the reviewer suggested.

We revised and extended the fourth paragraph in section 2.2 as 'Several studies conducted in forested environments indicated that OH measurements by Laser-Induced Fluorescence technique using wavelength modulation method might suffer from unknown internal-produced interference (Mao et al., 2012;Novelli et al., 2017), and the magnitude of interference is highly dependent on the specific design of the instrument, the operating parameters, and the type of environment in which the instrument is deployed (Fuchs et al., 2016;Novelli et al., 2014;Woodward-Massey et al., 2020;Cho et al., 2021). To investigate the possible OH interference in this campaign, we performed an extended chemical modulation experiment on 7 June. During the experiment, a chemical modulation device consisting of a Teflon tube with an inner diameter of 1.0 cm and a length of 10 cm was placed on the top of the OH sampling nozzle. About 17 slpm (standard liter per minute) of ambient air was drawn through the tube by a blower, 1 slpm of which entered the fluorescence cell. Tests on the transimition efficiency of OH through the chemical modulation device showed that the signals differed by less than 7% with or without chemical modulation device, indicating

the losses of ambient OH to the chemical modulation device were insignificant. For ambient measurement application, either propane (a 12% mixture in nitrogen, 6 sccm) diluted in a carrier flow of pure nitrogen (200 sccm) or pure nitrogen (200 sccm) was injected into the center of the tube alternatively every 5 minutes via two oppositely posited needles at the entrance of Teflon tube. The ambient OH signal can be then deduced by differentiating the signals from adjacent measurement modes with and without propane injection. The amount of the scavenger added is typically selected to be sufficiently high for reacting with ambient OH but not in excess in case reacting with internal-produced OH, and thus, the scavenging efficiency is usually kept around 90%. Calibrations of OH sensitivity with and without propane injection showed the scavenging efficiency of OH was around 93% in this experiment, and the kinetic calculation indicated the added propane removed less than 0.7% of the internal-produced OH. Therefore, the real ambient OH concentration can be obtained by multiplying the differential OH signal by the scavenging efficiency and by the instrument sensitivity. More details about the prototype chemical-modulation reactor used with PKU-LIF and the calculation method can be seen in Tan et al. (2017).'

4. *Section 2.4: How were photolysis frequencies treated in the model? Was the model constrained with all measured photolysis rates?*

**Answer:**

The model was constrained with the measured photolysis frequencies $j(O^1D)$, $j(NO_2)$, $j(HONO)$, $j(H_2O_2)$, $j(HCHO)$, and $j(NO_3)$. The photolysis frequencies of other species were calculated with the function of solar zenith angle.

We revised and added sentences at the end of the first paragraph in section 2.4 as 'Briefly, observations of the photolysis frequencies $j(O^1D)$, $j(NO_2)$, $j(HONO)$, $j(H_2O_2)$, $j(HCHO)$, and $j(NO_3)$, $O_3$, NO, $NO_2$, CO, $CH_4$, $SO_2$, HONO, C2-C12 VOCs, and certain oxygenated VOCs such as HCHO, acetaldehyde, glyoxal and acetone as well as the meteorological parameters were used to constrain the model with a time resolution of 5 min. Photolysis frequencies of other species were calculated in the model using the following function of solar zenith angle ($\chi$) and scaled to the ratio of measured to calculated $j(NO_2)$ to represent the effect from clouds. :
$$J = l \times (\cos \chi)^m \times e^{-n \times \sec \chi} \qquad \text{(Eq. 1)}$$

where the optimal values of parameters *l*, *m*, and *n* for each photolysis frequency were adopted (Saunders et al., 2003).'

5. *How detailed is the a-pinene oxidation scheme in RACM2? It would be useful to reference the a-pinene oxidation mechanism that was used.*

**Answer:**

We added a Table in the supplement to describe the $\alpha$-pinene oxidation scheme in RACM2.

**Table S1. $\alpha$-pinene oxidation mechanism in RACM2**

| Number | Reaction |
|--------|----------|
| #1 | API + OH → APIP |
| #2 | API + O$_3$ → 0.85 × OH + 0.1 × HO$_2$ + 0.2 × ETHP + 0.42 × KETP + 0.14 × CO + 0.02 × H$_2$O$_2$ + 0.65 × ALD + 0.53 × KET |
| #3 | API + NO$_3$ → 0.1 × OLNN + 0.9 × OLND |
| #4 | APIP + NO → 0.82 × HO$_2$ + 0.82 × NO$_2$ + 0.23 × HCHO + 0.43 × ALD + 0.44 × KET + 0.07 × ORA1 + 0.18 × ONIT |
| #5 | APIP + HO$_2$ → OP2 |
| #6 | APIP + MO$_2$ → HO$_2$ + 0.75 × HCHO + 0.75 × ALD + 0.75 × KET + 0.25 × MOH + 0.25 × ROH |
| #7 | APIP + ACO$_3$ → 0.5 × HO$_2$ + 0.5 × MO$_2$ + ALD + KET + ORA2 |
| #8 | APIP + NO$_3$ → HO$_2$ + NO$_2$ + ALD + KET |

API denotes $\alpha$-pinene; APIP denotes peroxy radicals formed from API; ETHP denotes peroxy radicals formed from ethane; KETP denotes peroxy radicals formed from ketone; ALD denotes C3 and higher aldehydes; KET denotes ketones; OLNN denotes NO$_3$-alkene adduct reacting to form carbonitrates and HO$_2$; OLND denotes NO$_3$-alkene adduct reacting via decomposition; ACT denotes acetone; ORA1 denotes formic acid; ONIT denotes organic nitrate; OP2 denotes higher organic peroxides; MO$_2$ denotes methyl peroxy radical; MOH denotes methanol; ROH denotes C3 and higher alcohols; ACO$_3$ denotes acetyl peroxy radicals; ORA2 denotes acetic acid and higher acids.

6. *The authors state that a first-order loss term equal to 8 hrs gave an observed to modelled ratio of 1.09 for PAN. Other box modelling studies, however, have had to impose a boundary layer height dependent loss rate to reproduce the diurnal trends observed for model-generated intermediates (e.g. Whalley et al., ACP, 21, 2125 – 2147, 2021). How well did the model predict the diurnal variation of PAN/ other model-generated species such as formaldehyde and glyoxal? How sensitive was the model-predicted OH and HO2 concentrations to the imposed loss rate?*

**Answer:**

We added some sentences to clarify the effect of the imposed loss rate on the model-generated intermediate in section 2.4. "The observed-to-model ratio of PAN concentration was 1.09 using this physical loss rate, while the modelled PAN concentration agreed to measurements from late morning to the midnight but slightly lower than measurements in the early morning (Fig. S2), which may relate to the effect of boundary layer height variation. To test the influence of boundary layer height diurnal variation, we performed a sensitivity test by imposing a boundary layer height (BLH, reanalysis data from European Centre for Medium-Range Weather Forecasts) dependent loss rate to all species. The model continuously underpredicted the

concentration in the early morning, and additionally, the model overestimated the observed PAN in the midday and afternoon (Fig. S2). This is because the boundary layer height dependent loss rate is largest at night, which makes the loss of PAN greater and further worsens the measurement-model comparison. Therefore, the treatment of a first-order loss term equal to 8 hours to all species in the model may not reflect the loss due to deposition but give a reasonable approximation on the overall physical loss of the model-generated intermediates. Nevertheless, the modelled OH and $HO_2$ concentrations were insensitive to the imposed loss rate (Fig. S3). The concentrations differed less than 0.5% between two cases for both OH and $HO_2$."

7. *Section 3.3: The modelled breakdown of OH reactivity and a comparison to the observed total reactivity was presented in Whalley et al., ACP, 21, 2125 – 2147, 2021 for the Beijing campaign and it would be useful to compare the modelled reactivity from EXPLORE-YRD to this. In Beijing a significant missing reactivity was determined and so the impact missing reactivity may have on the modelled radical concentrations during EXPLORE-YRD should be evaluated.*

**Answer:**

We added some discussions about the breakdown of OH reactivity in this study and the comparison to the observed total reactivity in other studies in Section 3.3.

[revised manuscript text omitted]

However, cares have to be taken if there was missing OH reactivity. As discussed in Sect. 4.2.1, a series of sensitivity tests had been performed to test the effect of missing reactivity on the modelled radical concentrations (Fig. S7). It turned out that when OH converted to $MO_2$, the modelled $HO_2$ increased by $6.2 \times 10^7$ $cm^{-3}$ compared to the base case. Incorporation of $HO_2$ heterogeneous uptake process upon this case would decrease the $HO_2$ concentration by 21% and 9%, respectively, for the coefficient of 0.2 and 0.08. Still, the measured and modelled $HO_2$ discrepancy (41%) would be beyond the uncertainty of $HO_2$ simulation for coefficient of 0.2. On the contrary, for cases that OH converted to ETEP and $ACO_3$, the modelled $HO_2$ decreased by $1.3 \times 10^7$ $cm^{-3}$ and $1.5 \times 10^7$ $cm^{-3}$, respectively compared to the base cases, possibly due to the faster radical termination rates through $RO_2 + HO_2$ in both these cases compared to that of $MO_2$. Nevertheless, the model sensitivity tests suggested that $HO_2$ uptake coefficient should be less than 0.2, if the $HO_2$ heterogeneous loss played a role during this campaign.'

8. *Pg 10, line 252: How sensitive is the model-generated contribution to OH reactivity to the physical loss rate imposed?*

**Answer:**

As shown in Fig. S3, the model-generated contribution to OH reactivity would change (increase in the afternoon and decrease at night) by less than 1 $s^{-1}$ if imposing a boundary layer height dependent loss rate rather than a constant first-order loss rate equals to a lifetime of 8 hours to all species. The daytime OH reactivity would increase by less than 5% compared to the base case, and the modelled OH and $HO_2$ concentration would change by less than 0.5%. It indicated the model-generated contribution to OH reactivity as well as OH and $HO_2$ concentrations were insensitive to the imposed physical loss rate. Moreover, as mentioned above, the performance of the model to predict PAN concentration was good if imposing boundary layer height dependent loss rate.

We added sentences to explain the effect of the imposed physical loss rate on the modelled OH reactivity in Section 3.3 as 'The model-generated OVOCs made comparable contribution to the measured ones (22% vs. 18%), and the model-generated contribution to OH reactivity was insensitive to the imposed physical loss rate (Fig. S3).'

9. *Section 4.2.1, lines 318 – 319: The good agreement between modelled and observed median HO2 should be discussed in light of possible missing OH reactivity (which, if caused by missing VOCs, could act as a source of modelled RO2 and HO2). This could also be discussed in section 4.2.3 which shows that the model to measured agreement for HO2 is reduced when loss to aerosols is considered.*

**Answer:**

We performed a series of sensitivity tests and added some discussion about the impact that missing OH reactivity might have on the modelled $HO_2$ radical concentrations in section 4.2.3.

'The agreements between measurement and model calculation of OH and $HO_2$ indicated that the base model without heterogeneous reaction captured the key processes for OH and $HO_2$ radical chemistry in this study.

As discussed in Sect. 4.2.1, a series of sensitivity tests had been performed to test the effect of missing reactivity on the modelled radical concentrations (Fig. S7). It turned out that when OH converted to $MO_2$, the modelled $HO_2$ would increase by $6.2\times10^7$ cm$^{-3}$ compared to the base case which makes more potential for the $HO_2$ heterogeneous loss. However, considering the potential effect of missing reactivity on $HO_2$, the measured and modelled $HO_2$ discrepancy (41%) would still be beyond the uncertainty of $HO_2$ simulation for coefficient of 0.2. On the contrary, for cases that OH converted to ETEP and ACO3, the modelled $HO_2$ decreased by $1.3\times10^7$ cm$^{-3}$ and $1.5\times10^7$ cm$^{-3}$, respectively compared to the base cases, possibly due to the faster radical termination rates through $RO_2+HO_2$ in both these cases compared to that of $MO_2$. Nevertheless, the model sensitivity tests suggested that HO2 uptake coefficient should be less than 0.2, if the HO2 heterogeneous loss played a role during this campaign.'

*10. Pg 13, line 335: Can the authors provide the % of OH that is recycled from isoprene via the H-shift mechanism?*

**Answer:**

We added the percentage of OH that is recycled from isoprene via the H-shift mechanism in Section 4.2.1 as 'However, during this campaign, isoprene concentration was only 0.2 ppb, contributing 5% of the modelled OH reactivity. The H-shift mechanism of isoprene derived peroxy radicals only increased 1.2% of the modelled OH concentration and thus play a minor role in OH chemistry. Therefore, other processes should account for the OH underestimation in low NO conditions.'

*11. Pg 14, lines 368 – 370: I disagree with this statement. Firstly, OH is underestimated by the model. Secondly, there are no OH reactivity observations or RO2 observations to test the model against. A much fuller discussion on the a-pinene oxidation mechanism used in this work is needed (see my later comment).*

**Answer:**

We revised these sentences as 'Other studies conducted in forested environments with a strong influence of monoterpenes from pine trees emission found discrepancies of up to three times in $HO_2$ measurement-model comparison (Kim et al., 2013;Wolfe et al., 2014;Hens et al., 2014). In present study, however, $HO_2$ concentration was well

reproduced by chemical model within combined uncertainty during daytime with high monoterpenes concentrations. Nevertheless, we cannot draw solid conclusion that the monoterpenes oxidation chemistry in environment with both strong anthropogenic and biogenic influences can be captured by the applied chemical mechanisms with respect to $HO_x$ concentration, since missing $HO_2$ sources and sinks might exist simultaneously but cancel out each other. Given that there were no OH reactivity or $RO_2$ observations in this study, we cannot rule out these possibilities.'

We added the a-pinene oxidation mechanism used in this work in Table S1, and performed a sensitivity test to test the influence of the isomerization of a-pinene derived RO species on modelled radical concentration in Section 4.2.2 (see the later answer).

*12. Pg 18, lines 455 – 457: The lower P(Ox) determined from modelled peroxy radicals relative to the observed demonstrates that the model under-predicts the observed HO2 concentration at NO concentrations greater than 1 ppb. This finding has been observed in a number of previous urban radical measurement campaigns and some discussion of this finding should be presented in this manuscript (perhaps in section 4.2).*

**Answer:**

We added some discussion about the underestimation of ozone production in urban atmosphere in Section 4.3.

'The discrepancy for observation and model derived P(Ox) mainly appears at NO concentration larger than 1 ppb (Fig. 9). This behavior has been observed in a number of previous urban radical measurement campaigns (Kanaya et al., 2008;Kanaya et al., 2012;Martinez, 2003;Ren et al., 2003;Ren et al., 2013;Elshorbany et al., 2012;Brune et al., 2016;Whalley et al., 2018;Tan et al., 2017), which was caused by the model underprediction of the observed $HO_2$ concentrations under high NO concentration (typically NO greater than 1 ppb). Although some of the previous $HO_2$ measurement might suffer from unrecognized interference from $RO_2$ species, this kind of interference have been minimized by lowering down the added NO concentration in recent studies (Griffith et al., 2016;Brune et al., 2016). However, the underestimation of ozone production from HO2 radical persist, indicating that the photochemical production mechanism of ozone under polluted urban environment is still not well understood.'

*13. Pg 18, lines 459 – 472: Whalley et al., ACP, 21, 2125 – 2147, 2021 highlighted that large RO2 species, such as those deriving from a-pinene, form RO species upon reaction with NO and these RO species can isomerise to form another RO2 species rather than forming HO2 directly. Are these types of RO isomerisations considered in the RACM2 mechanism? A discussion of the a-pinene oxidation mechanism used and the impact this may have on the calculated ozone production rate is warranted here.*

**Answer:**

As suggested in Whalley et al. (2021), the $RO_2$ species derived from α-pinene with ozone reaction, form RO upon reaction with NO, can isomerize to form another $RO_2$ rather than $HO_2$ directly, we performed a sensitivity test to substitute the reactions of α-pinene with ozone in RACM2 by those considering RO isomerization in MCM3.3.1 to investigate the impact of this mechanism may have on the modelled radical concentration and the calculated ozone production rate. It turned out to be that the isomerization mechanism had little influence on the modelled radical concentrations as well as on the calculated ozone production. Including the isomerization mechanism would decrease OH and $HO_2$ concentrations by $2.0 \times 10^4$ $cm^{-3}$ and $2.5 \times 10^7$ $cm^{-3}$, respectively, and reduce daily ozone production by 1 ppb.

We added some discussions about the impact of RO isomerization on modelled radical concentrations and ozone production in Section 4.2.2 and 4.3.

[revised manuscript text omitted]

---

## Author Response (AR1)

**Response to Community comment of Christa Fittschen to "OH and HO₂ radical chemistry at a suburban site during the EXPLORE-YRD campaign in 2018"**

**General comments:**

*It is interesting to notice that you observe the same increasing disagreement between modeled and measured OH concentration with decreasing NO concentrations. You say that the disagreement is probably not due to interference in the FAGE system, because using the chemical modulation system did not show any significant interference. Unfortunately, these experiments were carried out on June 7, which, looking at Figure 3, was a day where measurements and model were in very good agreement. So, it seems that no strong conclusion on absence of interferences in your FAGE system can be drawn from these observations.*

*It might have escaped from your attention that our group has identified a new OH interference in our FAGE system which would be able to explain such increasing disagreements between measured and modeled OH concentrations with decreasing NO: the product of the reaction between RO2 and OH radicals, trioxides ROOOH, leads in our FAGE system unequivocally to an OH signal1. Even though the lifetime of such trioxides and with this their absolute concentration is not known, this interference has the characteristics needed to explain your observation: the turnover of the reaction of RO2 + OH (and thus the ROOOH concentration) increases with decreasing NO concentration.*

*Even though not clearly stated, but I guess the reactions of RO2 + OH are not included in your reaction mechanism. This class of reaction has now been studied several timesf.e.2-3 and it is admitted that the rate constants are fast. Your Figure 8 allows to make a rough estimation on the importance of this class of reactions under your conditions: in the afternoon, the RO2 loss is dominated (»50 %) by their reaction with HO2. Taking the rate constant of RO2 + HO2 as »10-11 cm3s-1 and the rate constant of RO2 + OH as 10 times faster   (»10-10 cm3s-1), one can estimate that with 100 times less OH compared to HO2 (Figure 4), around 5% of the RO2 radicals will be lost through reaction with OH. It is admitted that the reaction of large peroxy radicals with OH leads nearly exclusively to the formation of ROOOH4, it can therefore be supposed that a non-negligible steady-state trioxide concentration can build up.*

*From these rough estimations it seems interesting to update your model by integration of RO2 + OH, add a reasonable loss process for the trioxides and check for correlation of trioxide concentration and degree of disagreement between measurement and model.*

**Answer:**

According to Christa's suggestions, we tested the importance of ROOOH on the degree of disagreement between measurement and model in this study by integrating the reactions of OH with all $RO_2$ species except for $CH_3O_2$ with a rate constant of $1.5×10^{-10}$ $cm^3$ $s^{-1}$ in our model, and adding a loss process for ROOOH with a rate of $10^{-4}$ $s^{-1}$,

corresponding to a lifetime of around 3 hours as suggested by Fittschen (2019);Fittschen et al. (2019), to represent an upper limit of ROOOH formation.

As shown in Fig. S8, the modelled ROOOH concentration peaked around 14:00 with the peak values on the order of $10^9 \sim 10^{10}$ cm$^{-3}$, which were comparable to the modelled values by Fittschen et al. (2019). However, the correlation of modelled ROOOH with the ratio of observed to modelled OH and the difference between observed and modelled OH showed little relevance ($R^2$=0.01 and 0.03, respectively on daytime basis, Fig. S9). It indicated that the ROOOH interference was not able to explain the disagreement between measurement and model in this study. Besides, the modelled ROOOH concentrations were high on the day when the chemical modulation tests were performed (7th June, Fig. S8). If ROOOH did cause significant OH interference in our FAGE system, it would also cause significant OH interference on that day, which however, was not discovered in the chemical modulation tests. It was also worth noting that incorporating the additional reactions of $RO_2$+OH into the existed reaction mechanism had little influence on the modelled OH, $HO_2$, and $RO_2$ concentrations. Nevertheless, ROOOH could be an OH interference in FAGE system, but as pointed also by Fittschen et al. (2019), the occurrence of this interference is highly dependent on the design and measurement conditions of different FAGE instruments. This interference was not the reason for the disagreement between model and measurement in our FAGE system at least for this campaign.

We added some discussions about the influence of the possible OH interference from ROOOH in Section 4.2.1 as 'Previous studies proposed that stabilized Criegee intermediates (SCIs) produced from reaction of ozone with alkenes and trioxides (ROOOH) produced from reaction of larger $RO_2$ with OH might cause artificial OH signals using LIF techniques (Novelli et al., 2017;Fittschen et al., 2019). However, chemical modulation tests on an ozone polluted day when both $O_3$ and ROOOH (modelled) concentrations were high (7 June) indicated insignificant interference for OH measurement in this study (Fig. S8). Furthermore, little relevance of ROOOH and the degree of disagreement between measurement and model was found in this study (Fig. S9), and thus, there is no hint for significant OH measurement interference during the EXPLORE-YRD campaign. However, one should note that the precision is not good enough to rule out the possibility.'

**Table S1. $\alpha$-pinene oxidation mechanism in RACM2**

| Number | Reaction |
|--------|----------|
| #1 | $API + OH \rightarrow APIP$ |
| #2 | $API + O_3 \rightarrow 0.85 \times OH + 0.1 \times HO_2 + 0.2 \times ETHP + 0.42 \times KETP + 0.14 \times CO$ $+ 0.02 \times H_2O_2 + 0.65 \times ALD + 0.53 \times KET$ |
| #3 | $API + NO_3 \rightarrow 0.1 \times OLNN + 0.9 \times OLND$ |
| #4 | $APIP + NO \rightarrow 0.82 \times HO_2 + 0.82 \times NO_2 + 0.23 \times HCHO + 0.43 \times ALD$ $+ 0.44 \times KET + 0.07 \times ORA1 + 0.18 \times ONIT$ |
| #5 | $APIP + HO_2 \rightarrow OP2$ |
| #6 | $APIP + MO_2 \rightarrow HO_2 + 0.75 \times HCHO + 0.75 \times ALD + 0.75 \times KET + 0.25 \times MOH$ $+ 0.25 \times ROH$ |
| #7 | $APIP + ACO_3 \rightarrow 0.5 \times HO_2 + 0.5 \times MO_2 + ALD + KET + ORA2$ |
| #8 | $APIP + NO_3 \rightarrow HO_2 + NO_2 + ALD + KET$ |

API denotes $\alpha$-pinene; APIP denotes peroxy radicals formed from API; ETHP denotes peroxy radicals formed from ethane; KETP denotes peroxy radicals formed from ketone; ALD denotes C3 and higher aldehydes; KET denotes ketones; OLNN denotes $NO_3$-alkene adduct reacting to form carbonitrates and $HO_2$; OLND denotes $NO_3$-alkene adduct reacting via decomposition; ACT denotes acetone; ORA1 denotes formic acid; ONIT denotes organic nitrate; OP2 denotes higher organic peroxides; $MO_2$ denotes methyl peroxy radical; MOH denotes methanol; ROH denotes C3 and higher alcohols; $ACO_3$ denotes acetyl peroxy radicals; ORA2 denotes acetic acid and higher acids.

6. *The authors state that a first-order loss term equal to 8 hrs gave an observed to modelled ratio of 1.09 for PAN. Other box modelling studies, however, have had to impose a boundary layer height dependent loss rate to reproduce the diurnal trends observed for model-generated intermediates (e.g. Whalley et al., ACP, 21, 2125 – 2147, 2021). How well did the model predict the diurnal variation of PAN/ other model-generated species such as formaldehyde and glyoxal? How sensitive was the model-predicted OH and HO2 concentrations to the imposed loss rate?*

**Answer:**

We added some sentences to clarify the effect of the imposed loss rate on the modelgenerated intermediate in section 2.4. "The observed-to-model ratio of PAN concentration was 1.09 using this physical loss rate, while the modelled PAN concentration agreed to measurements from late morning to the midnight but slightly lower than measurements in the early morning (Fig. S2), which may relate to the effect of boundary layer height variation. To test the influence of boundary layer height diurnal variation, we performed a sensitivity test by imposing a boundary layer height (BLH, reanalysis data from European Centre for Medium-Range Weather Forecasts) dependent loss rate to all species. The model continuously underpredicted the concentration in the early morning, and additionally, the model overestimated the observed PAN in the midday and afternoon (Fig. S2). This is because the boundary layer height dependent loss rate is largest at night, which makes the loss of PAN greater and further worsens the measurement-model comparison. Therefore, the treatment of a first-order loss term equal to 8 hours to all species in the model may not reflect the loss due to deposition but give a reasonable approximation on the overall physical loss of the model-generated intermediates. Nevertheless, the modelled OH and $HO_2$ concentrations were insensitive to the imposed loss rate (Fig. S3). The concentrations differed less than 0.5% between two cases for both OH and $HO_2$."

7. *Section 3.3: The modelled breakdown of OH reactivity and a comparison to the observed total reactivity was presented in Whalley et al., ACP, 21, 2125 – 2147, 2021 for the Beijing campaign and it would be useful to compare the modelled reactivity from EXPLORE-YRD to this. In Beijing a significant missing reactivity was determined and so the impact missing reactivity may have on the modelled radical concentrations during EXPLORE-YRD should be evaluated.*

**Answer:**

We added some discussions about the breakdown of OH reactivity in this study and the comparison to the observed total reactivity in other studies in Section 3.3.

[revised manuscript text omitted]

Section 4.2.3: 'The agreements between measurement and model calculation of OH and $HO_2$ indicated that the base model without heterogeneous reaction captured the key processes for OH and $HO_2$ radical chemistry in this study.
However, cares have to be taken if there was missing OH reactivity. As discussed in Sect. 4.2.1, a series of sensitivity tests had been performed to test the effect of missing reactivity on the modelled radical concentrations (Fig. S7). It turned out that when OH converted to $MO_2$, the modelled $HO_2$ increased by $6.2{\times}10^7$ cm$^{-3}$ compared to the base case. Incorporation of $HO_2$ heterogeneous uptake process upon this case would decrease the $HO_2$ concentration by 21% and 9%, respectively, for the coefficient of 0.2 and 0.08. Still, the measured and modelled $HO_2$ discrepancy (41%) would be beyond the uncertainty of $HO_2$ simulation for coefficient of 0.2. On the contrary, for cases that OH converted to ETEP and $ACO_3$, the modelled $HO_2$ decreased by $1.3{\times}10^7$ cm$^{-3}$ and $1.5{\times}10^7$ cm$^{-3}$, respectively compared to the base cases, possibly due to the faster radical termination rates through $RO_2+HO_2$ in both these cases compared to that of $MO_2$. Nevertheless, the model sensitivity tests suggested that $HO_2$ uptake coefficient should be less than 0.2, if the $HO_2$ heterogeneous loss played a role during this campaign.'

8. *Pg 10, line 252: How sensitive is the model-generated contribution to OH reactivity to the physical loss rate imposed?*

**Answer:**

As shown in Fig. S3, the model-generated contribution to OH reactivity would change (increase in the afternoon and decrease at night) by less than 1 s$^{-1}$ if imposing a boundary layer height dependent loss rate rather than a constant first-order loss rate equals to a lifetime of 8 hours to all species. The daytime OH reactivity would increase by less than 5% compared to the base case, and the modelled OH and $HO_2$ concentration would change by less than 0.5%. It indicated the model-generated contribution to OH reactivity as well as OH and $HO_2$ concentrations were insensitive to the imposed physical loss rate. Moreover, as mentioned above, the performance of the model to predict PAN concentration was good if imposing boundary layer height dependent loss rate.

We added sentences to explain the effect of the imposed physical loss rate on the modelled OH reactivity in Section 3.3 as 'The model-generated OVOCs made comparable contribution to the measured ones (22% vs. 18%), and the model-generated

contribution to OH reactivity was insensitive to the imposed physical loss rate (Fig. S3).'

9. *Section 4.2.1, lines 318 – 319: The good agreement between modelled and observed median HO2 should be discussed in light of possible missing OH reactivity (which, if caused by missing VOCs, could act as a source of modelled RO2 and HO2). This could also be discussed in section 4.2.3 which shows that the model to measured agreement for HO2 is reduced when loss to aerosols is considered.*

**Answer:**

We performed a series of sensitivity tests and added some discussion about the impact that missing OH reactivity might have on the modelled $HO_2$ radical concentrations in section 4.2.3.

'The agreements between measurement and model calculation of OH and $HO_2$ indicated that the base model without heterogeneous reaction captured the key processes for OH and $HO_2$ radical chemistry in this study.

As discussed in Sect. 4.2.1, a series of sensitivity tests had been performed to test the effect of missing reactivity on the modelled radical concentrations (Fig. S7). It turned out that when OH converted to $MO_2$, the modelled $HO_2$ would increase by $6.2 \times 10^7$ cm$^{-3}$ compared to the base case which makes more potential for the $HO_2$ heterogeneous loss. However, considering the potential effect of missing reactivity on $HO_2$, the measured and modelled $HO_2$ discrepancy (41%) would still be beyond the uncertainty of $HO_2$ simulation for coefficient of 0.2. On the contrary, for cases that OH converted to ETEP and ACO3, the modelled $HO_2$ decreased by $1.3 \times 10^7$ cm$^{-3}$ and $1.5 \times 10^7$ cm$^{-3}$, respectively compared to the base cases, possibly due to the faster radical termination rates through $RO_2 + HO_2$ in both these cases compared to that of $MO_2$. Nevertheless, the model sensitivity tests suggested that HO2 uptake coefficient should be less than 0.2, if the HO2 heterogeneous loss played a role during this campaign.'

10. *Pg 13, line 335: Can the authors provide the % of OH that is recycled from isoprene via the H-shift mechanism?*

**Answer:**

We added the percentage of OH that is recycled from isoprene via the H-shift mechanism in Section 4.2.1 as 'However, during this campaign, isoprene concentration was only 0.2 ppb, contributing 5% of the modelled OH reactivity. The H-shift mechanism of isoprene derived peroxy radicals only increased 1.2% of the modelled OH concentration and thus play a minor role in OH chemistry. Therefore, other processes should account for the OH underestimation in low NO conditions.'

11. *Pg 14, lines 368 – 370: I disagree with this statement. Firstly, OH is underestimated by the model. Secondly, there are no OH reactivity observations or RO2*

*observations to test the model against. A much fuller discussion on the a-pinene oxidation mechanism used in this work is needed (see my later comment).*

**Answer:**

We revised these sentences as 'Other studies conducted in forested environments with a strong influence of monoterpenes from pine trees emission found discrepancies of up to three times in $HO_2$ measurement-model comparison (Kim et al., 2013;Wolfe et al., 2014;Hens et al., 2014). In present study, however, $HO_2$ concentration was well reproduced by chemical model within combined uncertainty during daytime with high monoterpenes concentrations. Nevertheless, we cannot draw solid conclusion that the monoterpenes oxidation chemistry in environment with both strong anthropogenic and biogenic influences can be captured by the applied chemical mechanisms with respect to $HO_x$ concentration, since missing $HO_2$ sources and sinks might exist simultaneously but cancel out each other. Given that there were no OH reactivity or $RO_2$ observations in this study, we cannot rule out these possibilities.'

We added the a-pinene oxidation mechanism used in this work in Table S1, and performed a sensitivity test to test the influence of the isomerization of a-pinene derived RO species on modelled radical concentration in Section 4.2.2 (see the later answer).

*12. Pg 18, lines 455 – 457: The lower P(Ox) determined from modelled peroxy radicals relative to the observed demonstrates that the model under-predicts the observed HO2 concentration at NO concentrations greater than 1 ppb. This finding has been observed in a number of previous urban radical measurement campaigns and some discussion of this finding should be presented in this manuscript (perhaps in section 4.2).*

**Answer:**

We added some discussion about the underestimation of ozone production in urban atmosphere in Section 4.3.

'The discrepancy for observation and model derived P(Ox) mainly appears at NO concentration larger than 1 ppb (Fig. 9). This behavior has been observed in a number of previous urban radical measurement campaigns (Kanaya et al., 2008;Kanaya et al., 2012;Martinez, 2003;Ren et al., 2003a;Ren et al., 2013;Elshorbany et al., 2012;Brune et al., 2016;Whalley et al., 2018;Tan et al., 2017), which was caused by the model underprediction of the observed $HO_2$ concentrations under high NO concentration (typically NO greater than 1 ppb). Although some of the previous $HO_2$ measurement might suffer from unrecognized interference from $RO_2$ species, this kind of interference have been minimized by lowering down the added NO concentration in recent studies (Griffith et al., 2016;Brune et al., 2016). However, the underestimation of ozone production from HO2 radical persist, indicating that the photochemical production mechanism of ozone under polluted urban environment is still not well understood.'

*13. Pg 18, lines 459 – 472: Whalley et al., ACP, 21, 2125 – 2147, 2021 highlighted that large RO2 species, such as those deriving from a-pinene, form RO species upon reaction with NO and these RO species can isomerise to form another RO2 species rather than forming HO2 directly. Are these types of RO isomerisations considered in the RACM2 mechanism? A discussion of the a-pinene oxidation mechanism used and the impact this may have on the calculated ozone production rate is warranted here.*

**Answer:**

As suggested in Whalley et al. (2021), the $RO_2$ species derived from α-pinene with ozone reaction, form RO upon reaction with NO, can isomerize to form another $RO_2$ rather than $HO_2$ directly, we performed a sensitivity test to substitute the reactions of α-pinene with ozone in RACM2 by those considering RO isomerization in MCM3.3.1 to investigate the impact of this mechanism may have on the modelled radical concentration and the calculated ozone production rate. It turned out to be that the isomerization mechanism had little influence on the modelled radical concentrations as well as on the calculated ozone production. Including the isomerization mechanism would decrease OH and $HO_2$ concentrations by $2.0 \times 10^4$ cm$^{-3}$ and $2.5 \times 10^7$ cm$^{-3}$, respectively, and reduce daily ozone production by 1 ppb.

We added some discussions about the impact of RO isomerization on modelled radical concentrations and ozone production in Section 4.2.2 and 4.3.

[revised manuscript text omitted]

**General comments:**

*Ma et al. presented the OH and HO$_2$ radical measurement conducted at a rural site in Yangtze River Delta, China. A box model based on RACM2-LIM1 was used to simulate the radicals concentrations, which was underestimated the OH at low NO (<1ppb) conditions. The influence of monoterpene oxidation and heterogenous loss of HO$_2$ on aerosol surface were tested. The authors also summarized the HOx measurement in (sub)urban environments all over the world, which gave a very nice overview of the HOx studies. Overall, the manuscript is well-written, and the investigation is scientifically sound. I strongly recommend the publication with some minor modification.*

**Answer:**

We would like to thank the reviewer for comments and questions which helped us to improve the manuscript. The reviewer comments are given below together with our responses and changes made to the manuscript. The technical comments were changed accordingly and not listed below for simplify.

**Major comments:**

1. *Line 189, Information on back trajectory analysis of air masses is better added in the SI for the demonstration.*

**Answer:**

We added the back trajectory analysis of air masses during this campaign in SI (Fig. S3).

2. *Figure 2: Oxidation product of HCHO and Gly diurnal pattern show peaks in morning, which is not the common sense of their formation pathway in summertime. Even for the urban site in YRD, these two species also show comparable abundances but higher levels around noontime (Guo et al., 2021). Could the authors explain more about the anthropogenic emission-related origin?*

**Answer:**

HCHO and glyoxal are not only photochemical product of VOC oxidation, but also from primary biogenic and anthropogenic emissions, including vegetation, biomass burning, fossil fuel combustion, and biofuel consumption. Since the EXPLORE-YRD campaign was conducted during a harvest season, biomass burning events occurred frequently. The elevated concentration might be due to the biomass burning activity.

We added some sentences to explain this in Sect. 3.1 as 'Since this campaign was conducted during a harvest season, agriculture biomass burning might be responsible for the elevated HCHO and glyoxal in the early morning (Guo et al., 2021;Liu et al., 2020;Wang et al., 2017;Silva et al., 2018).'

3. *The investigation of possible influence of monoterpene on radical chemistry and ozone production is interesting and of importance. I agree with Reviewer#1 that additional discussion on the monoterpene oxidation (most probably using alpha-pinene as a representative) would benefit the manuscript. However, lacking the simultaneous measurement of $RO_2$ concentration and OH reactivity, one could not really draw solid conclusion from this comparison. It's unlikely to test the chemical mechanism with the current dataset. Therefore, I suggest to minimize the discussion on monoterpene and only show the potential impact. The same applies to the closure of OH reactivity.*

**Answer:**

We agree with the reviewer that lacking of the simultaneous measurement of $RO_2$ concentration and OH reactivity, we could not draw solid conclusion. Nevertheless, we performed a series of sensitivity tests to investigate the impact of the applied monoterpene oxidation mechanism and the potential missing OH reactivity on the modelled radical concentrations and ozone production rates in this study. Detailed discussion can be found in the response to Reviewer#1.

4. *I would like to suggest the authors can review other literatures reporting the precursors related species, e.g. HONO, HCHO etc to the ROx production in YRD areas to strengthen the discussion, and destruction also.*

**Answer:**

We added some discussions of the radical sources and destructions in YRD region in Section 4.1. as 'Two recent winter campaign in the same region also found HONO photolysis dominated radical primary source, contributing 38% to 53% of the total radical sources, despite the overall radical production rates were several times lower than that in summertime (Lou et al., 2022;Zhang et al., 2022). The photolysis of HONO is one of the most important radical primary sources in worldwide urban and suburban areas for both summer (Ren et al., 2003b;Dusanter et al., 2009;Michoud et al., 2012;Whalley et al., 2018;Tan et al., 2017) and winter time (Ren et al., 2006;Kanaya et al., 2007;Kim et al., 2014;Tan et al., 2018;Ma et al., 2019).'

5. *In addition, comparison with wintertime OH and HOx in YRD may benefit the discussion on measurements and simulation throughout the manuscript, which can help to give the full view of the year for YRD region, e.g. Zhang et al., 2022.*

**Answer:**

[revised manuscript text omitted]

---

## Author Response (AR2)

1. The authors have not provided the answer to the following initial query in their responses:

   From laboratory tests, what is the conversion efficiency of an alkene $RO_2$ to OH at 5 ppm and 2.5 ppm NO? Furthermore, if the $HO_2$ signal measured with the high NO injection is 6% higher than the HO2 signal that is measured with the low NO injection, then I think that this does indicate that there is an $RO_2$ interference being observed. I suggest the authors attempt to put an upper limit on the $RO_2$ interference using the methodology described in Whalley et al., (AMT, 2013) (equations 7 and 8).

   **Answer:**

   Previous laboratory experiments of another LIF system (FZJ-LIF) with the same cell design and operating parameters to PKU-LIF indicated the conversion efficiency of isoprene derived $RO_2$ to OH was lower than 0.1 at NO concentration of $4\times10^{12}$ cm$^{-3}$ (Fuchs et al., 2011) (Figure 5, 0.4 mm nozzle). Therefore, the conversion efficiencies for the NO concentrations of 5 ppm and 2.5 ppm used in this study (i.e. $5\times10^{11}$ cm$^{-3}$ and $2.5\times10^{11}$ cm$^{-3}$ at cell pressure of 4 hPa) were expected to be less than 0.1.
   According to Equation 7 and 8 in Whalley et al. (2013), the $HO_2$ interference from $RO_2$ radical can be calculated by multiplying the complex $RO_2$ concentrations ($RO_2i$) with corresponding conversion efficiency ($\alpha$). Unfortunately, $RO_2$ was not measured during this campaign while one would expect a strong correlation between $RO_2$ (or $RO_2i$) and $HO_2$. Previous summer campaigns in China demonstrated that the $RO_2i$ to $HO_2$ ratio varies from 0.6 in a rural site in Wangdu (Tan et al., 2017) to 2 in an urban site in Beijing (Whalley et al., 2021). As the chemical condition encountered in YRD was more similar to that of Wangdu (the Beijing campaign was conducted at an urban site), it was reasonable to assume the $RO_2i$ to $HO_2$ ratio in this study was closer to 0.6. By applying the conversion efficiency of 0.1 as an upper limit and assuming $RO_2i$ to $HO_2$ ratio to be 0.6, the maximum $HO_2$ interference from $RO_2$ radicals should be closer to 6% of the $HO_2$ measurement in this study. But we acknowledge that the interference would increase, if $RO_2i$ to $HO_2$ ratio become larger.

   We revised the paragraph as 'To minimize the potential interference from $RO_2$, the added NO mixing ratio was switched between 2.5 ppm and 5 ppm every 2 minutes, corresponding to the $HO_2$ conversion efficiencies of 10% and 20%, respectively. The expected $RO_2$ conversion efficiency for both modes was below 10% for this experimental setup for isoprene derived $RO_2$ from laboratory tests (Fuchs et al. 2011). The extent of the $RO_2$-interference was also proportional to the complex-$RO_2$-to-$HO_2$ ratio. Unfortunately, $RO_2$ was not measured during this campaign but one would expect a strong correlation between $RO_2$ (or complex-$RO_2$) and $HO_2$ (Tan et al., 2017; Whalley et al., 2021). Previous field summer campaigns in China showed that, the ratio of complex-$RO_2$ to $HO_2$ varies from 0.6 at a rural site in Wangdu (Tan et al., 2017) to 2 at an urban site in Beijing (Whalley et al., 2021). As

the chemical condition encountered in YRD was more similar to that of Wangdu (the Beijing campaign was conducted at an urban site), it was reasonable to assume the complex-$RO_2$ to $HO_2$ ratio in this study was closer to 0.6. Therefore, by applying the $RO_2$ conversion efficiency of 0.1 as an upper limit, the maximum $HO_2$ interference from $RO_2$ radicals should be closer to 6% of the $HO_2$ measurement in this study assuming complex-$RO_2$ to $HO_2$ ratio to be 0.6.'

2. The authors have not responded fully to the following:
   How well did the model predict the diurnal variation of PAN/ other model-generated species such as formaldehyde and glyoxal? Could the authors include the modelled and observed formaldehyde and glyoxal profiles in the SI and with modelled and observed PAN?

   **Answer:**

   The diurnal variations of modelled and observed PAN, formaldehyde and glyoxal were shown in Fig. S2. As discussed in the last version of revision, the modelled PAN concentration agreed to measurements from late morning to the midnight but slightly lower than measurements in the early morning, and the discrepancy was not caused by the effect of boundary layer height variation. While the model under-predicted the HCHO concentrations and over-predicted the glyoxal concentrations, which might be related to the significant primary emission of HCHO and missing sinks of glyoxal in the current mechanisms. However, the missing sources and sinks of HCHO and glyoxal are not the scope of this study. To avoid interruption from incapability of model performance, both HCHO and glyoxal were constrained to observations in this study.

   We added some discussions about the observed and modelled HCHO and glyoxal in the model description section as 'In addition, sensitivity test without HCHO and glyoxal constrained indicated that model would under-predicted the HCHO and over-predicted the glyoxal concentrations (Fig. S2), which might be related to the significant primary emission of HCHO and missing sinks of glyoxal in the current mechanisms. However, the missing sources and sinks of HCHO and glyoxal are not the scope of this study. To avoid interruption from incapability of model performance, both HCHO and glyoxal were constrained to observations in this study.'

[Figure]

Figure S2. The mean diurnal profiles of measured and modelled PAN (a), HCHO (b), and glyoxal (c) concentrations. (a) The base model run (Base) applied a first-order loss term equivalent to a lifetime of 8 hours to all species. The other model run (w. BLH var) imposed a boundary layer height (BLH, derived from ECMWF) dependent loss rate to all species. (b) and (c) The model run (Mod) free the HCHO and glyoxal compared to the base model run (Base) in (a). The grey areas denote nighttime.

**References**

Fuchs, H., Bohn, B., Hofzumahaus, A., Holland, F., Lu, K. D., Nehr, S., Rohrer, F., and Wahner, A.: Detection of HO2 by laser-induced fluorescence: calibration and interferences from RO2 radicals, Atmospheric Measurement Techniques, 4, 1209-1225, 10.5194/amt-4-1209-2011, 2011.

Tan, Z., Fuchs, H., Lu, K., Hofzumahaus, A., Bohn, B., Broch, S., Dong, H., Gomm, S., Haeseler, R., He, L., Holland, F., Li, X., Liu, Y., Lu, S., Rohrer, F., Shao, M., Wang, B., Wang, M., Wu, Y., Zeng, L., Zhang, Y., Wahner, A., and Zhang, Y.: Radical chemistry at a rural site (Wangdu) in the North China Plain: observation and model calculations of OH, HO2 and RO2 radicals, Atmospheric Chemistry and Physics, 17, 663-690, 10.5194/acp-17-663-2017, 2017.

Whalley, L. K., Blitz, M. A., Desservettaz, M., Seakins, P. W., and Heard, D. E.: Reporting the sensitivity of laser-induced fluorescence instruments used for HO2 detection to an interference from RO2 radicals and introducing a novel approach that enables HO2 and certain RO2 types to be selectively measured, Atmospheric Measurement Techniques, 6, 3425-3440, 10.5194/amt-6-3425-2013, 2013.

Whalley, L. K., Slater, E. J., Woodward-Massey, R., Ye, C. X., Lee, J. D., Squires, F., Hopkins, J. R., Dunmore, R. E., Shaw, M., Hamilton, J. F., Lewis, A. C., Mehra, A., Worrall, S. D., Bacak, A., Bannan, T. J., Coe, H., Percival, C. J., Ouyang, B., Jones, R. L., Crilley, L. R., Kramer, L. J., Bloss, W. J., Vu, T., Kotthaus, S., Grimmond, S., Sun, Y. L., Xu, W. Q., Yue, S. Y., Ren, L. J., Acton, W. J. F., Hewitt, C. N., Wang, X. M., Fu, P. Q., and Heard, D. E.: Evaluating the sensitivity of radical chemistry and ozone formation to ambient VOCs and NOx in Beijing, Atmospheric Chemistry and Physics, 21, 2125-2147, 10.5194/acp-21-2125-2021, 2021.